# Multi-modal quantification of pathway activity with MAYA

Yuna Landais[1] & Céline Vallot ⬡[2,3,4] ✉

Signaling pathways can be activated through various cascades of genes depending on cell identity and biological context. Single-cell atlases now provide the opportunity to inspect such complexity in health and disease. Yet, existing reference tools for pathway scoring resume activity of each pathway to one unique common metric across cell types. Here, we present MAYA, a computational method that enables the automatic detection and scoring of the diverse modes of activation of biological pathways across cell populations. MAYA improves the granularity of pathway analysis by detecting subgroups of genes within reference pathways, each characteristic of a cell population and how it activates a pathway. Using multiple single-cell datasets, we demonstrate the biological relevance of identified modes of activation, the robustness of MAYA to noisy pathway lists and batch effect. MAYA can also predict cell types starting from lists of reference markers in a cluster-free manner. Finally, we show that MAYA reveals common modes of pathway activation in tumor cells across patients, opening the perspective to discover shared therapeutic vulnerabilities.

The identification of cell type and function is the driving force of most single-cell studies. Such approaches are based on lists of canonical marker genes and pathway databases. Standard scRNA-seq analysis pipelines involve steps of dimensionality reduction and clustering before starting any marker or pathway analysis[1–3], which makes the resulting conclusions highly dependent on the chosen algorithm and clustering parameters. In the case of cancer datasets, such clustering-based approaches appear inadequate to identify shared transcriptional programs across tumors as cancer cells tend to cluster per patient[4–9] rather than group by biological similarities. Several approaches have emerged, bypassing dimensionality reduction and clustering, and proposing to score pathway activity directly for individual cells rather than for clusters. Such pooling of several gene-based measurements into scores has proven extremely powerful for the interpretation of sparse and noisy scRNA-seq datasets[10,11]. A recent benchmark[12] presented Pagoda2[13] and AUCell[14] as two of the top performing tools for pathway activity scoring. They are based on different scoring methods—AUCell estimates the proportion of highly expressed genes in each pathway while Pagoda2 uses the weights of the first principal component from Principal Component Analysis (PCA)—and each proposes a way to select significant scores. Nonetheless, both tools compute a unique activity score per pathway for all cells, implying that genes of a given signaling pathway should have coordinated expression across cell types.

Biological evaluation of pathway activation and more recently single-cell studies have repeatedly demonstrated the heterogeneity of cell functions depending on the biological context. Yet most single-cell studies analyze pathway activation with single scores based on gene lists extracted from bulk data. Such curated gene lists represent the current reference biological knowledge, that the community uses to make biological sense of sparse and noisy scRNA-seq data. Adding more specialized curated gene lists to databases—detailing cellular functions according to cell identity—is ongoing but it will take some time to be completed. To inspect existing pathway databases with single-cell resolution, we developed MAYA (Multimodes of pathwAY Activation), a tool that detects—for each pathway—the different modes of activation across cell types, each mode relying on a different subset of genes. We argue that MAYA could be a way for currently available

[1]One Biosciences, Paris, France. [2]CNRS UMR3244, Institut Curie, PSL University, Paris, France. [3]Translational Research Department, Institut Curie, PSL University, Paris, France. [4]Single Cell Initiative, Institut Curie, PSL University, Paris, France. ✉e-mail: celine.vallot@onebiosciences.fr

biological knowledge to meet the granularity reached by single-cell data and help researchers go deeper in their understanding of complex cellular mechanisms. Particularly, in the case of cancer datasets, we show that MAYA can detect cell type specific modes of pathway activation for both the microenvironment and tumor cells, defining common expression programs across patients, in line with recently identified tumor "meta-programs"[15].

## Results

### MAYA method

MAYA enables comprehensive pathway study thanks to multimodal scoring of gene lists in individual cells (Fig. 1). Provided a scRNA-seq count matrix and pathway lists, MAYA detects all biologically relevant ways to activate each pathway based on subgroups of genes and summarizes their activity in each cell in a multimodal score matrix (Fig. 1a). This activity matrix can then be used to identify groups of cells sharing similar activation of provided pathways and to visualize datasets in lower dimensions (Fig. 1b). As a comparison, reference tools that measure pathway activity, such as AUCell[14] or Pagoda2[13], provide a unique activity score per pathway where MAYA can provide several.

MAYA is built on two main functions that are applied to each provided gene list: the detection of activation modes and the selection of biologically relevant ones. Detection of modes is performed thanks to a PCA on a normalized gene-cell expression matrix restricted to pathway genes (Fig. 1a). The purpose of such decomposition of the matrix is to find, within the pathway, genes which expression is coordinated and variable across cells, and to score their activity in individual cells. Each principal component (PC) represents a possible mode of activation of the pathway, that is characterized by the genes that contribute the most to the PC, and by a score that corresponds to the cell coordinate on the PC. Each gene can contribute to several PCs and therefore to several modes.

However, not all detected modes reflect a relevant biological pattern in the data and some could be driven by outliers, either cells and/or genes; this probability increases as modes explain less and less variance in the dataset (Supplementary Fig. 1a). We thus developed a method to assess the informativity of each mode, based on two biologically interpretable criteria. First, an informative mode should be more active in a minimal subset of cells compared with other cells. This is assessed by detecting bimodal distributions of scores across cells and checking that the group of active cells represents more than a minimum fraction of the population. This fraction can be determined based on previous knowledge of the underlying biology or set arbitrarily (Supplementary Fig. 1b–d). Second, an informative mode should be driven by enough genes to be considered as a mode of activation per se and not solely correspond to the expression of a single outlier gene. To that end, we determined a cutoff for maximal variance of each gene of a mode, indicative of how much a gene can contribute on its

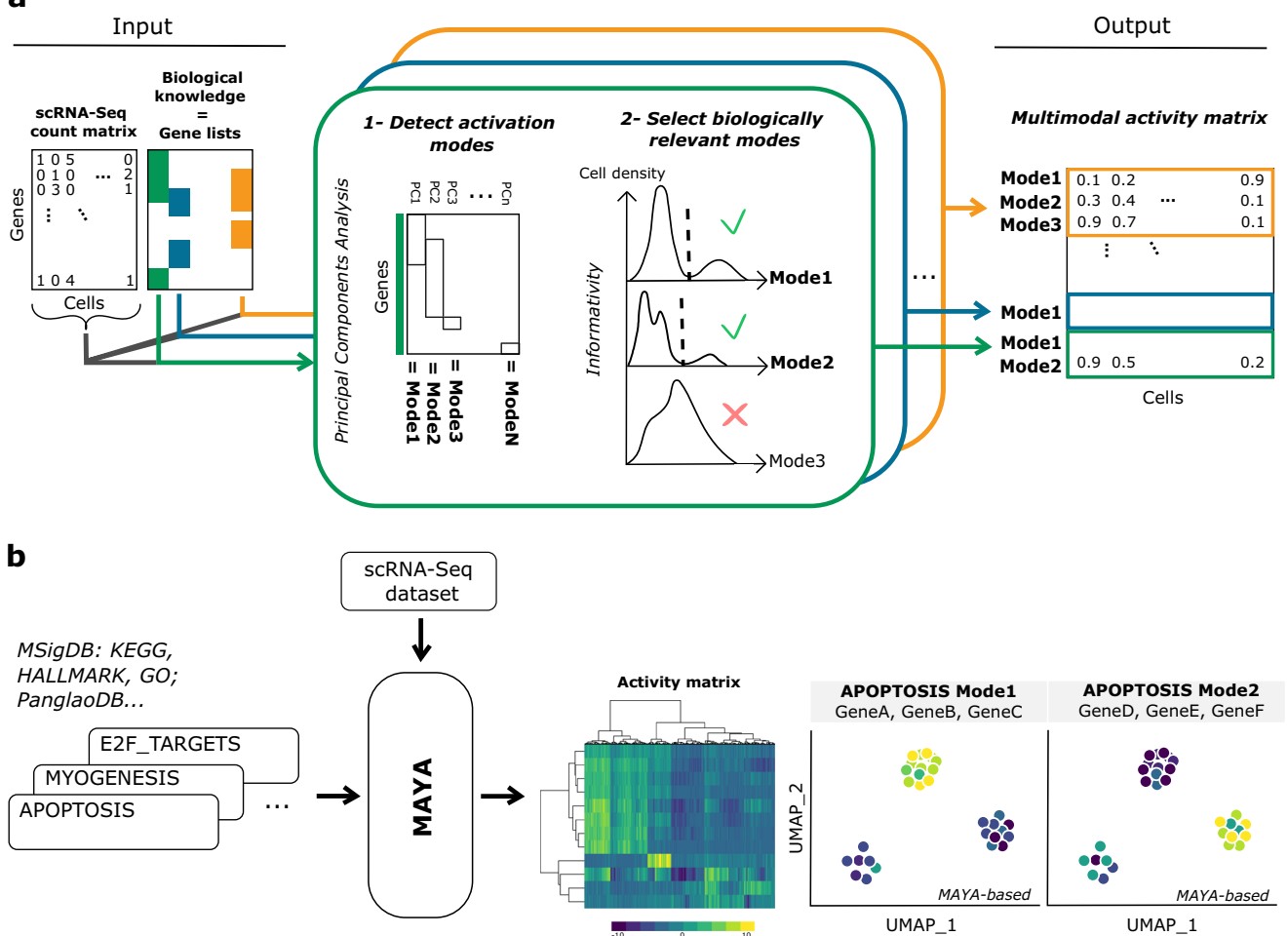

**Fig. 1 | MAYA overview. a** MAYA takes as input a scRNA-Seq dataset and reference gene lists, and produces as output an activity matrix, with for each cell its activity score for each mode of every reference gene list. **b** Example of MAYA outputs: a heatmap to visualize the modes of activation of reference pathways, or a Uniform Manifold Approximation and Projection (UMAP) of the activity matrix to visualize cells according to any annotation (activity scores for different modes, predicted cell type or any user annotation).

own (Supplementary Fig. 1e, f). Default cutoff value was chosen to maximize the number of modes detected as informative while keeping a high average number of genes significantly contributing to each mode (Supplementary Fig. 1g). This method is robust across single-cell technologies, whether 10X Chromium or Smart-Seq2, as shown with a PBMC dataset generated using both technologies[16] (Supplementary Fig. 1h).

Although MAYA's main purpose is to detect multimodal activation of pathways, it can also perform unimodal activity scoring to detect cell identity from any cell marker lists. To this end, we have developed a built-in function to leverage MAYA's scoring and informativity methods to annotate cells in a dataset. This approach is based on activation of the first mode using PanglaoDB[17] as input gene lists by default (Methods). This function allows cluster-free cell type annotation in a timely fashion as it annotates a dataset of around 16,000 cells in less than 1 min and 125,000 cells in ~15 min (Supplementary Fig. 1i).

## MAYA detects biologically relevant multimodal pathway activity in kidney

The main distinguishing feature of MAYA over existing pathway scoring tools is the multimodality of its activity score, which proves useful when studying broad pathways in complex biological systems. We first sought to demonstrate its ability to detect cell-type specific activation modes of hallmark pathways. For that, we ran MAYA on a dataset of normal kidney and immune cells from Young et al.[18], from which we selected cells from five distinct subtypes for clarity ($n = 1252$). We used the MSigDB Hallmark pathways[19] as input gene lists, covering main biological functions. Unsupervised clustering on the multimodal activity matrix shows MAYA detects modes that distinguish different cell populations (Fig. 2a). More specifically, we noticed that modes from the same pathway were specifically activated in different cell types. As an example, the Allograft rejection pathway presents two modes of activation (Fig. 2b–d): (i) mode 1, driven by the expression of *CTSS* and *SPI1*—known to have a critical role in antigen presentation[20] and gene regulation during myeloid development[21]—and specific to monocytes (specificity of 0.57), and (ii) mode 2, driven by *CD2*, *CD3E* and *CD3D*—coding for T cell surface proteins—and by *CD8A* and *CD8B* - coding for the CD8 antigen—and specific to CD8 T cells (specificity of 0.88). In contrast, AUCell and Pagoda2 both describe this pathway with a single score, corresponding to an aggregation of MAYA's mode 1 and 2, or mode 1 only respectively (Fig. 2e). Another detailed example is shown in Supplementary Fig. 2 for the TNFA signaling via NFKB pathway, where four activation modes were detected with MAYA based on their bimodal activity distribution (Supplementary Fig. 2a): one specific to monocytes, one to CD8 T cells and two to endothelial cells (Supplementary Fig. 2b–d). Interestingly, each mode involves a different interleukin, each specific to the population in which the mode is found to be active: (i) IL6ST is a signal transducer, which dimerizes with IL6R and is bound for instance by IL-6, resulting in the activation of downstream cascades in endothelial cells[22], (ii) IL1B is a lymphocyte activating factor produced by monocytes, macrophages and neutrophils, and (iii) IL7R is associated with T cell differentiation. Altogether, we demonstrate here that MAYA identifies relevant cell-type specific modes of pathway activation from broad reference gene lists.

To test both the stability and the ability of MAYA to detect biologically relevant signal from noisy gene lists, we added 10, 50, 100, and 200 random genes to the initial 200 genes of the pathways Allograft rejection and TNFA signaling via NFKB; each experiment was repeated 100 times. For the Allograft Rejection pathway, the two initial activation modes were detected for all modified gene lists with a high cell-type specificity, whatever the level of added noise (Fig. 2f, g). These results also show the accuracy of our selection method to detect relevant modes, as we rarely detect additional activation modes (corresponding to PC3/mode 3) even when randomly increasing the reference gene lists. Similarly, for the TNFA signaling pathway, the first three modes are

robust to noise, with a decrease in sensitivity of detection when adding more than 100 unrelated genes (Supplementary Fig. 2e).

## MAYA detects biologically relevant multimodal pathway activity in colon

We then illustrated the relevance of the biological insight gained by using multimodal pathway analysis for another tissue with a dataset of colon and immune cells from Lee et al.[23]—from which we selected cells from ten distinct cell types ($n = 1415$)—and using the MSigDB KEGG and REACTOME pathways[24]. Both analyses recover cell-type specific activation modes, given the clustering of cells by cell type on the heatmaps derived from the activity matrix (Supplementary Fig. 3a, c). Focusing on KEGG cell adhesion molecules list, we observed that MAYA was able to detect several well-known types of cell-cell adhesion processes starting from the mixed general reference list (Fig. 3a, b and Supplementary Fig. 3b): (i) mode 1 driven by the expression of HLA genes coding MHC class II molecules[25], detected in antigen-presenting cells—monocytes and dendritic cells—and B cells, with a specificity of 0.29, 0.27 and 0.15 respectively, (ii) mode 2 driven by the expression of genes coding for claudins and cadherins located at tight junctions[26,27], specifically activated in epithelial cells (specificity of 0.24 and 0.16 for enterocytes and goblet cells respectively), and (iii) mode 3 driven by the expression of T cell membrane molecules, specific to Regulatory T cells (specificity of 0.29).

Applying MAYA to the REACTOME pathway ion channel transport, we were able to detect different types of ion channels and functions, specific to each cell population (Fig. 3c, d and Supplementary Fig. 3d). Mode 1 is specific to colon epithelial cells (specificity of 0.34 and 0.24 for enterocytes and goblet cells respectively, Fig. 3d) and corresponds to two types of ion channels—Epithelial Sodium Channel (ENaCs) and Na,K-ATPase[28] - that have been shown to participate to the regulation of salt and water absorption from the colon lumen[29,30]. In particular, activation mode 1 captures genes regulating ENaCs and their residence at the apical membrane: *SCNN1A* encodes a subunit of ENaCs[31], NEDD4L participates to ENaCs ubiquitination which leads to their retrieval from cell surface[32] and SGK1 is known to phosphorylate NEDD4L product, which decreases its binding to ENaCs[33,34]. Mode 4 is specific to goblet cells only, driven by the expression of the genes *CLCA1* and *BEST2*. These two genes are associated with Calcium-activated Chloride Channels (CaCCs) that have been shown to participate in epithelial secretion[35]. Mode 3 is specific to pericytes and smooth muscle cells (specificity of 0.16 and 0.22 respectively) and is associated with Calcium homeostasis (*ATP2B4*, *PLN*, *CASQ2*) and Na,K-ATPases (*FXYD1*, *FXYD6*, *ATP1A2*, *ATP1B2*), two important channels for the membrane polarization of contractile cells. Finally, mode 2, mainly active in monocytes and dendritic cells (specificity of 0.33 and 0.16 respectively), involves genes associated with acidification of intracellular organelles through colocalization of V-type proton ATPases[36] (*ATP6V1B2*, *ATP6AP1*, *ATP6V1F*, *ATP6V0E1*, *ATP6V0D2*[37]) and Chloride channels[38] (*TTYH3*, *CLIC2*), a process necessary for phagocytosis. Altogether, as for the kidney, starting from broad reference databases, MAYA untangles pathway activities specific to each cell type, revealing precise cell functions.

## MAYA assigns cell identity

We then leveraged MAYA's scoring and selection ability to robustly assign cell identity. We applied MAYA to PanglaoDB cell marker lists and the subsets of kidney and colon datasets used previously (Fig. 4a, d). We demonstrate that MAYA enabled an assisted and accurate annotation of each cell in the two datasets, using the initial cell type annotation by authors as a reference (Fig. 4b, e). We compared the accuracy of our predictions with the ones obtained with two other cell type identification methods: Cell-ID[39], based on Multiple Correspondence Analysis (MCA), and SCINA[40], based on an expectation-maximization model. MAYA presents among the highest rates of recall and precision for both datasets (Fig. 4c, f and Supplementary

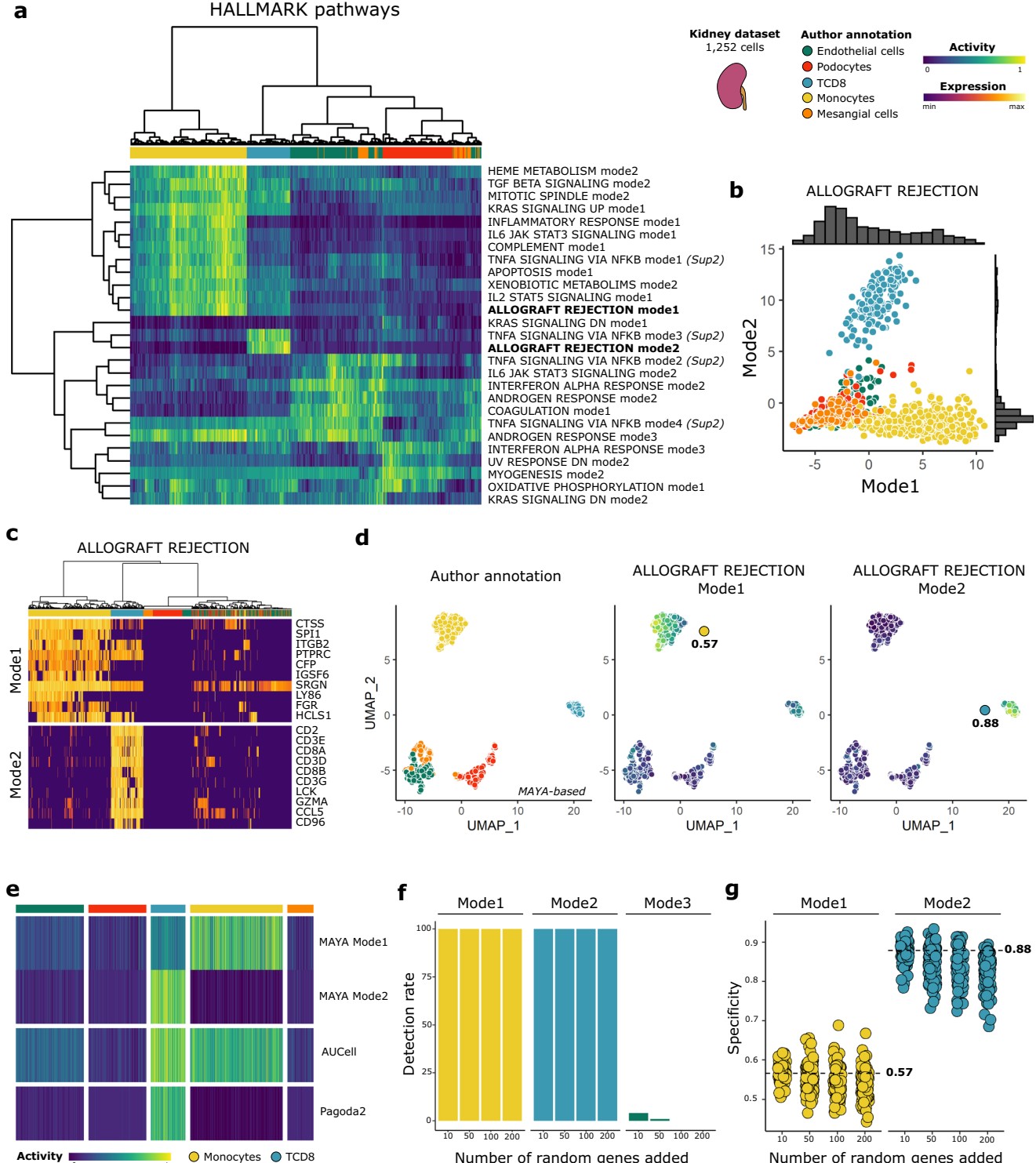

**Fig. 2 | Activation modes of Hallmark pathways in kidney with MAYA.**
**a** Heatmap of activity matrix computed on kidney dataset with MSigDB Hallmark pathways, initial author annotation is indicated above heatmap. The two activation modes of Allograft Rejection are highlighted in bold and further described in the subsequent panels, and the four modes of activation of TNFA signaling via NFKB are further described in Supplementary Fig. 2. **b** Scatterplot of Mode 2 versus Mode 1 cell activity scores. Associated density histograms are indicated on the sides of the graph. **c** Heatmap of scaled gene expression for top 10 contributing genes for Mode1 (top) and Mode2 (bottom) of Allograft Rejection pathway, ordered by decreasing contribution for each. **d** UMAP representation of activity matrix of Hallmark pathways, cells are colored according to author annotation, or activity scores of modes 1 and 2 of Allograft rejection pathway. Specificity score of cell populations is displayed next to relevant clusters. **e** Heatmap of activity scores computed by Pagoda2, AUCell and MAYA for Allograft Rejection pathway, cells are grouped according to author annotation. **f** Barplot representation of the detection rate of modes 1 to 3 for the pathway Allograft Rejection when adding various numbers of random genes to the pathway gene list ($n = 100$ experiments each). Barplots are colored according to the cell population with the highest specificity score for the identified mode. **g** Jitter representation of specificity scores of modes 1 and 2 grouped by level of added noise, data points are colored according to author annotation. Specificity obtained for each mode with initial gene list is represented with a dashed line.

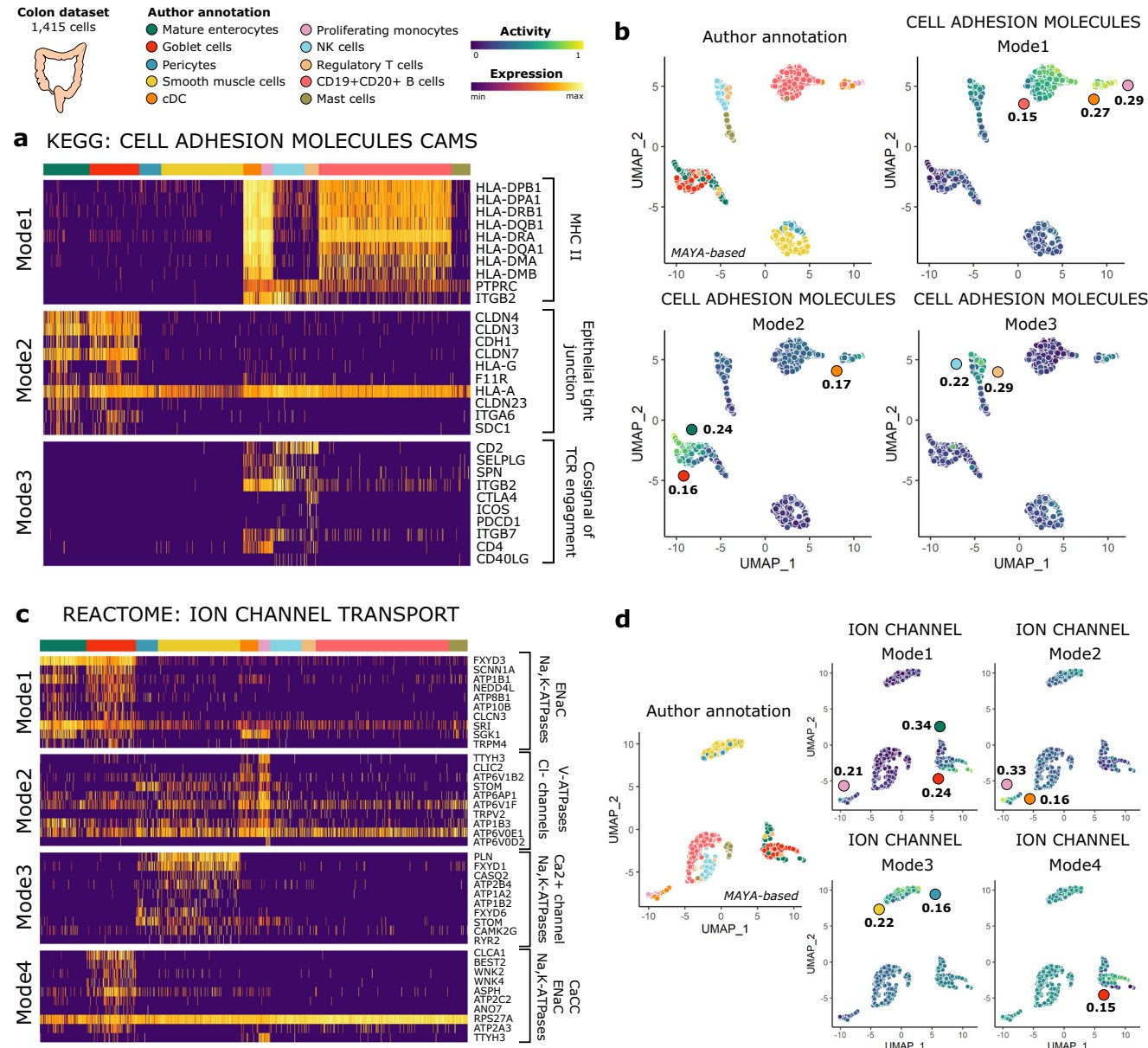

**Fig. 3 | KEGG and REACTOME activity in colon with MAYA. a** Heatmap of scaled gene expression for top10 contributing genes for the three activation modes of KEGG Cell Adhesion Molecules pathway, ordered by decreasing contribution. **b** UMAP representation of activity matrix of KEGG pathways, cells are colored according to author annotation, or activity scores of the three modes of Cell Adhesion Molecules pathway. Specificity score of cell populations is displayed next to relevant clusters. **c** Heatmap of scaled gene expression for top10 contributing genes for the four activation modes of the Ion Channel Transport pathway, ordered by decreasing contribution. **d** UMAP representation of activity matrix of REACTOME pathways, cells are colored according to author annotation, or activity scores of the four modes of Ion Channel Transport pathway. Specificity score of cell populations is displayed next to relevant clusters.

Fig. 4a, b). We finally tested the scalability of MAYA and its ability to detect rare cell types on a dataset with 16,815 cells from ovarian tumors[6] (Supplementary Fig. 4c). Overall, MAYA had an average precision of 51% and recall of 68%. Notably, B cells were identified with a precision and recall of 98% when they represent only 4.9% of the dataset and endothelial cells with a precision of 100% and recall of 85% when they represent 0.2% of cells in the dataset (Supplementary Fig. 4d). Lower precision is achieved for some types probably due to overlap between cell type markers in PanglaoDB, such as between NK cells and T cells (28 shared markers out of 80 and 95 markers respectively), dendritic cells and macrophages (34 shared out of 121 and 128 markers respectively), and endothelial cells and fibroblasts (13 shared out of 187 and 171 markers respectively). All three pairs of cell types share more genes than with any other type from the PanglaoDB lists.

Furthermore, as batch effect is a main concern in single-cell analyses, we tested whether MAYA was affected by such technical biases. We worked on a dataset containing $n = 5179$ cells from laryngeal squamous cell carcinoma biopsies of two patients with a batch effect between patients[41]. Using standard gene-based scRNA-seq matrix processing, cells from the same cell types – whether cells from the microenvironment or the tumors – indeed cluster by patient whereas clustering on the MAYA activity matrix groups cells by cell type, with cells from both patients within the same cluster (Fig. 4g). To quantify the inter-patient overlap between clusters of similar cell types, we computed the Shannon Diversity Index (SDI) for both methods as well as for clusters obtained with the reference integration tool Harmony[42] and an integration method based on Canonical Correlation Analysis[43] (CCA) (Supplementary Fig. 4e). MAYA had an average SDI of 0.77

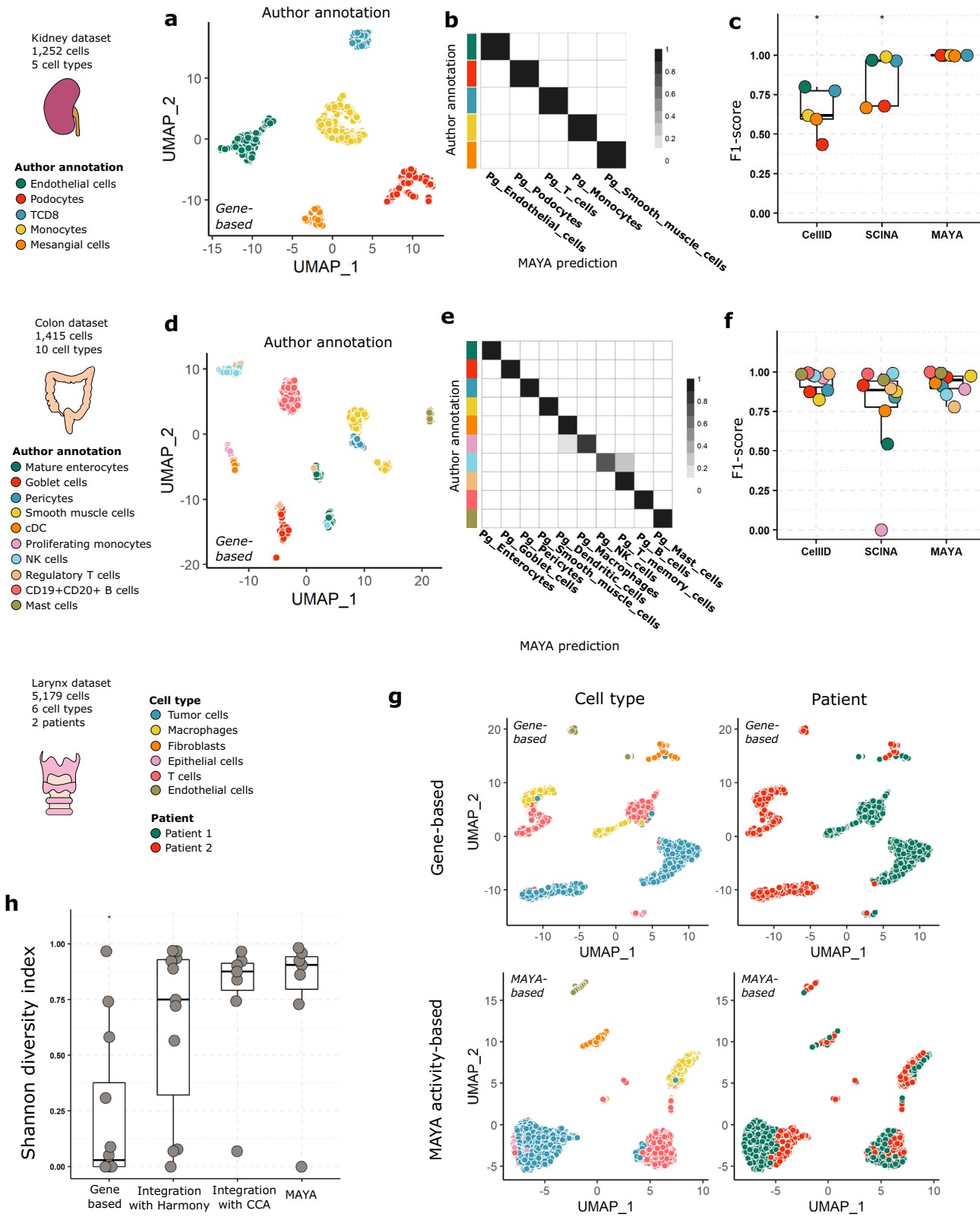

against 0.76, 0.63, and 0.23 for the integration based on Harmony, on CCA and the gene-based method respectively (Fig. 4h). To confirm these observations, we compared the four methods on a pancreas dataset containing $n = 14,890$ cells generated with five different single-cell technologies[44] (Supplementary Fig. 4e, f). Similarly, the gene-based

method generated technology-specific clusters whereas MAYA along with the two other integration methods consistently mixed cells from different technologies. In addition to pathway scoring, MAYA can perform accurate cell type annotation independently of batch effect, making it an all-in-one tool to address both cell identity and function.

**Fig. 4 | MAYA annotates cell type. a** Gene-based UMAP representation of kidney dataset, cells are colored according to author annotation. **b** Heatmap representing for each author annotation (rows) the fraction of cells labeled with each MAYA annotation (columns) for the kidney dataset. **c** Overlaid jitter and boxplot representation of F1-scores for automatic annotation of the kidney dataset using Cell-ID, SCINA and MAYA, data points are colored according to author annotation; $n = 5$ cell types; Center line, median; box limits, upper and lower quartiles; whiskers, 1.5× interquartile range; Adjusted $p$-values from two-sided Wilcoxon test are symbolized with: *: <0.05, **: <0.01, ***: <0.001, ****: <0.0001 (pval = 0.011 for Cell-ID and for SCINA). **d** Gene-based UMAP representation of colon dataset, cells are colored according to author annotation. **e** Heatmap representing for each author annotation (rows) the fraction of cells labeled with each MAYA annotation (columns) for the colon dataset. **f** Overlaid jitter and boxplot representation of F1-scores for automatic annotation of the colon dataset using Cell-ID, SCINA and MAYA, data points are colored according to author annotation; $n = 10$ cell types; Center line, median; box limits, upper and lower quartiles; whiskers, 1.5× interquartile range; Adjusted $p$-values from two-sided Wilcoxon test are not significant. **g** UMAP representation of the larynx dataset, either gene-based or based on activity matrix of PanglaoDB cell-type markers lists, cells are colored according to cell type or to patient. **h** Overlaid jitter and boxplot representation of Shannon Diversity Index (SDI), for clusters derived from gene-based dimensionality reduction, Harmony dimensionality reduction, CCA dimensionality reduction and MAYA activity matrix of the larynx dataset; $n = 12, 11, 7,$ and 7 independent clusters respectively; Center line, median; box limits, upper and lower quartiles; whiskers, 1.5× interquartile range; Adjusted $p$-values from two-sided Wilcoxon test are symbolized with: *: <0.05, **: <0.01, ***: <0.001, ****: <0.0001. Adjusted $p$-value is 0.026 for Gene-based relative to MAYA and non-significant for other comparisons.

## MAYA detects common modes of pathway activation across cancer patients

Patient-specificity of cancer cells is currently a major limitation for the comprehensive study of cancer scRNA-seq datasets. Cells of the microenvironment coming from different patients can easily group together, showing the absence of a major batch effect between samples, while tumor cells form distinct clusters[4–9]. Such behavior is thought to be due in part to the genetic variations across tumor cells from different patients, notably copy-number variations. Integration methods, correcting for general batch effect in samples, such as Harmony[42], are not suited to deal with such cell-type specific effect.

We demonstrate here that MAYA can be an alternative to gene-based or integration-based methods to identify common transcriptional features between cancer cells across patients. Using an ovarian cancer dataset, we show that MAYA identifies several modes of pathway activation shared across patients (Fig. 5a–c and Supplementary Fig. 5a, b) that are associated with known cancer hallmarks. Indeed, top specific modes of epithelial cancer cells reflect the expression of genes associated with early response to estrogen or the P53 pathway (specificity of 0.45 and 0.31 respectively), that relate to tumor growth and proliferation (Fig. 5c). Such tumor-cell specific activation would not have been found with a classical GSEA approach (Supplementary Fig. 5c). MAYA also identifies modes of pathway activation specific to the tumor microenvironment, e.g a cell-type specific activation of complement genes in macrophages (specificity of 0.24) and of angiogenesis in cancer-associated fibroblasts (CAFs) (specificity of 0.40).

MAYA's multimodality allows to untangle several cell-type specific modes of activation for biological phenomena that are commonly difficult to sort out between cell populations within the tumors and their microenvironment. For example, MAYA detects different modes of epithelial-to-mesenchymal transition (EMT) (Fig. 5d, e): mode 1 specific to CAFs/mesothelial cells (specificity of 0.47 and 0.36 respectively), mode 2 specific to tumor cells (specificity of 0.30) and mode 3 to macrophages (specificity of 0.19) (Fig. 5d). MAYA identifies a combination of genes that characterizes EMT occurring in epithelial cells, with *LAMA3* and *LAMC2* being exclusive to this cell type (Fig. 5d). These two genes expressed by basal epithelium code for two subunits of laminin 332, an essential component of epithelial basement membrane that promotes tumor cell motility[45,46]. In CAFs, MAYA detects EMT as driven mainly by genes encoding proteins from the extracellular matrix (ECM) including collagens, which have been shown to promote EMT in the tumor microenvironment directly[47] or by increasing the ECM stiffness[48,49]. A third mode of EMT, characterized by the expression of the gene SPP1, is found in macrophages; macrophages have indeed been shown to be involved in EMT induction in various types of cancer[50–53]. Two additional modes are detected but are not as cell-type specific as the others (Supplementary Fig. 5a, maximum specificity scores of 0.12). Interestingly, EMT modes specific to CAFs and to tumor cells were also detected in two other cancer

datasets (breast[54] and lung[55], Supplementary Fig. 5d). When evaluated in the same cells, the ovary, breast and lung modes display high pairwise Pearson correlation coefficients between datasets and high cell type specificity for both the CAF and tumor-specific modes (Supplementary Fig. 5e).

MAYA also identifies two different modes of activation of the early estrogen response (Fig. 5f and Supplementary Fig. 6a), one specific to tumor cells, and another specific to CAFs, consistent with the observation that CAFs can use ER-mediated signaling to promote tumor cell proliferation[56,57]. MAYA also helps to untangle the respective contribution of cancer cells and its microenvironment to the hemostatic imbalance observed in cancer[58,59]. It detects coagulation modes with high specificity for CAFs and mesothelial cells (0.31 and 0.32), tumor cells (0.22) and macrophages (0.24) (Fig. 5g and Supplementary Fig. 6b). In addition, when running MAYA with MSigDB KEGG gene lists, we detect two modes of activation of the WNT pathway, one specific to tumor cells and the other specific to CAFs (Fig. 5h). It is indeed known from the literature that Wnt signaling is a complex pathway involving β-catenin in its canonical form, but at least three other non-canonical Wnt-mediated pathways have been proposed to function independently of β-catenin[60]. One of them involves activation of calcium/calmodulin-dependent kinase II (CamKII), protein kinase C and phosphatase CaN (PPP3CC), and is referred to as the "Wnt Ca2+ pathway"[61]. Inspecting the top contributing genes of the MAYA modes, we observed that mode 1 contains solely genes from the canonical WNT pathway, driven potentially by the ligand WNT7A, while mode 2 contains genes implicated in both canonical and non-canonical Wnt Ca2+ pathway, in particular through the ligand WNT5A[62,63]. This example illustrates how MAYA can untangle different ways to activate the same biological function.

Finally, Gavish et al.[15] recently proposed a method based on Non-negative Matrix Factorization (NMF) to identify de novo tumor "meta-programs" shared across patients (see Methods). Applied to epithelial ovarian cancer cells (Supplementary Fig. 6c–e), this approach identifies 'meta-programs' related to functions that were also detected by MAYA (E2F targets, hypoxia) and others that were not found as specific to tumor cells by MAYA as TNFA signaling. On the other hand, no "meta-program" was found to be associated with estrogen response− in contrast to MAYA−as it is activated in all tumor cells, showing the complementarity between the two approaches, one able to detect shared activation programs across all cells or a subset, and the other focusing on intra-tumoral heterogeneity.

Altogether, MAYA appears extremely powerful to detect modes of pathway activation across tumor cells from different patients as well as within the microenvironment - novel combinations of genes within known global reference gene lists. We see with these examples that MAYA can discover refined gene lists, specific to each population, matching the biological interpretation of pathway activation to the granularity of the single-cell measurements. For the study of tumor

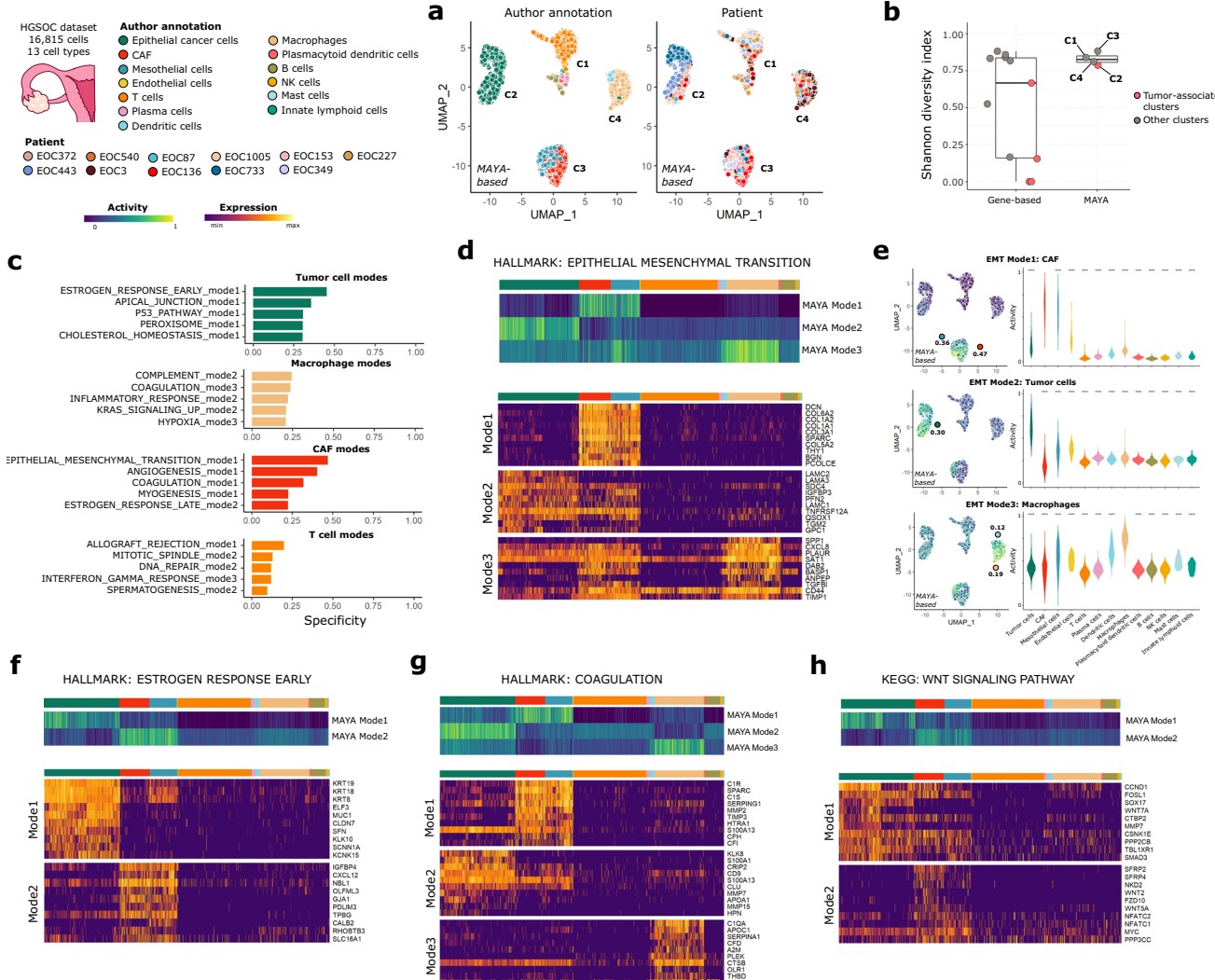

**Fig. 5 | MAYA detects pathway activation in tumors across patients. a** UMAP representation of activity matrix of Hallmark pathways, cells are colored according to author annotation and patient. Clusters derived from activity matrix are displayed next to relevant groups of cells. **b** Overlaid jitter and boxplot representation of Shannon Diversity Index (SDI), for clusters derived from gene-based dimensionality reduction and MAYA activity matrix of the ovary dataset. Clusters corresponding to tumor cells are colored in pink; n = 11 and 4 independent clusters respectively; Center line, median; box limits, upper and lower quartiles; whiskers, 1.5× interquartile range. **c** Barplot representation of specificity scores of the top5 specific modes for the four most prevalent populations in the dataset. **d** Heatmap of activity scores of the three modes of the Hallmark Epithelial Mesenchymal Transition (EMT) pathway, initial author annotation is indicated above heatmap. Heatmap of scaled gene expression for top10 contributing genes for the three modes of EMT, ordered by decreasing contribution. **e** UMAP representation of activity matrix of Hallmark pathways, cells are colored according to activity scores of the three EMT modes. Specificity score of cell populations is displayed next to relevant clusters. Violin plots of activity scores for corresponding modes, grouped by author annotation (adjusted p-values from two-sided Wilcoxon test are symbolized with: *: <0.05, **: <0.01, ***: <0.001, ****: <0.0001). Heatmap of the activity scores for the modes of (**f**) Hallmark Estrogen Response Early pathway, (**g**) Hallmark Coagulation pathway and (**h**) KEGG Wnt pathway, cells are grouped according to author annotation. Heatmap of scaled gene expression for top10 contributing genes for corresponding modes, ordered by decreasing contribution.

cells specifically, MAYA appears complementary to other recently proposed methods based on the discovery of de novo consensus transcriptomic programs.

## Discussion

MAYA sorts out the different modes of pathway activation specific to each cell type, by automatically detecting subgroups of genes within reference pathways, and computing several scores of pathway activation. We show that MAYA leverages existing biological knowledge to extract cell-type specific ways of activating pathways from single-cell datasets. In addition to pathway analysis, MAYA performs assisted cell typing as a side function, making it an all-in-one tool for both cell type and cell function identification. MAYA proves particularly useful for single-cell cancer datasets, by (i)

identifying common modes of pathway activation across patients in tumor cells, and also by (ii) dissecting the contribution of each population—fibroblast, immune & tumor cell—to the activation of a given pathway.

In comparison to previously published methods (AUCell[14], Pagoda2[13], ROMA[64], and UCell[65]), MAYA provides multiple activation scores per pathway. With MAYA, we simplified bimodal detection by focusing on inflection points and introducing two biologically interpretable parameters, easily tunable by users: (i) a minimum proportion of cells that should activate a mode for the mode to be considered relevant and (ii) a maximum contribution to a mode that a single gene can have. MAYA will detect activation modes for a pathway given that the provided datasets present both cells that activate and cells that do not activate such pathway.

We have also challenged the robustness to noise of our scoring and informativity methods and showed MAYA can detect relevant biological signal from noisy pathway lists. It can prove very useful as we know pathway and cell markers manual curation is very time-consuming. Here, we argue that MAYA can take as input non-curated and potentially very exhaustive pathway or cell type lists and detect biological signal if they contain any.

We also leveraged our methods of scoring and selection of informative scores to propose a built-in function to annotate cells using the first mode of activation of PanglaoDB cell type markers lists. This method has performance results equivalent to Cell-ID[39] and SCINA[40], two packages specialized in cell type annotation. MAYA is scalable to large datasets (>100,000 cells, in 15 min), unsensitive to batch effect, and is able to accurately detect and annotate cell populations representing less than 5% of cells. However, MAYA might not be suited to annotate cell types using few marker genes—we recommend using MAYA with lists containing at least ten genes. In addition, for cell types which share many markers in reference databases, users will need additional expertise to validate the assisted annotation.

Finally, MAYA appears particularly useful when studying single-cell datasets from cancer patients that do not suffer from batch effect on all cell types but from patient-specificity for tumor cells. There is currently no standard way to address this challenge for data interpretation and a growing need to understand common cancer features across patients. Recently, Gavish et al.[15] provided the community with recurrent shared transcriptional programs across patient and tumor types by describing 41 "meta-programs" grouped in 11 hallmarks of intra-tumor heterogeneity. These "meta-programs" were inferred de novo by studying scRNA-seq from multiple tissues and cancer types. This approach is very complementary to ours, where we interrogate existing knowledge instead of performing de novo identification. MAYA identifies common modes of activation across tumor cells, which could be compared to such tumor meta-programs. In addition, MAYA deciphers the respective contribution of each cell population to the activation of a given pathway, by defining the group of genes that drive the pathway activity in each contributing population. Both inter and intra-patient features of MAYA will enable the identification of shared therapeutic vulnerabilities across patients, as well as various strategies to target them within the tumor eco-system.

## Methods

### Matrix preprocessing
All count matrices were processed with Seurat v3 to get the gene-based cell embeddings and check the consistency of author's annotations. Matrices were log-normalized using scale factor 10,000. Top 2000 variable features were found using "vst" method. PCA and UMAP were computed with default settings, using first 10 PCs for UMAP, which constitutes the "gene-based UMAP". For the larynx dataset, the two datasets were read separately and merged in a unique Seurat object of 5179 cells. The authors did not provide their annotation, so we followed the default Seurat pipeline on each individual count matrix, performed PCA and default clustering. We then annotated clusters based on expression of cell type markers described in the publication.

### Detailed description of MAYA algorithm
**Building count matrix.** For a provided gene list, the log-normalized CPM matrix is subsetted to keep all cells but only genes from the list. Rows of the matrix are then scaled so that more highly expressed genes do not weight more than the others in the PCA that is later performed. The sign of each principal component is then chosen to favor the direction for which the absolute value of gene contribution is the highest. Each mode is scaled between 0 and 1. An iterative

process then begins: we evaluate the informativity of each successive PC starting from PC1. If a PC is found uninformative, the iteration stops, and we do not interrogate further PCs. There is however an exception for PC1: we interrogate PC2 even if PC1 is uninformative, as PC2 can still explain a significance part of the variance. The final activity matrix is built by gathering all modes from all gene lists in a single matrix with modes as rows and cells as columns.

**Informativity.** For each successive mode, a density curve is drawn from the distribution to get local maxima and minima. A bimodal curve is expected to have at least one minimum that will be low enough relative to its surrounding maxima on the y-axis to mark a clear distinction between two groups of cells (difference of at least 10% of global maximum density). Only local minima with abscissa superior to the one of the global maximum are considered and iteratively evaluated in decreasing order as the point is to detect extreme behaviors and activation patterns that potentially occur in rare populations. The iteration stops when a potential minimum meets the criteria, or none was found. As this process relies on the detection of inflection points that depends itself on the adjustment of the density curve to the distribution, we start with an adjustment meant to detect global variations of distributions and if none are detected we test a more fitted adjustment to ensure no significant local variation was missed. Then follow two additional checks to ensure the biological relevance of the detected mode. First, we filter out modes that are activated in very few cells as they could be outliers. The user can adjust this parameter based on what he expects to observe in the dataset or the number of cells from rarer cell type or set it to default 5%. The second biological check is based on the number of genes potentially contributing to the mode. However, it is hard to set a definition of what is a contributing gene to PCA; here we consider that contributing genes contribute more than they would be expected i.e., if all genes from the pathway contributed the same (1/number of genes in the pathway). Given that pathways have various sizes, it is difficult to set a hard cutoff on this number of genes contributing to the mode. Instead, we chose to set a cut-off on the maximum contribution of a gene to a mode. As the sum of squared gene contributions is equal to 1, if a gene contributes to up to 0.8, there is not much contribution left for other genes to share and this mode is probably driven by this unique gene. As a mode should represent joint expression of groups of genes, we do not consider these monogenic modes biologically significant. Setting a threshold of 0.4 allows to remove monogenic modes while keeping a relatively large number of modes with higher cell type specificity. This parameter can also be changed by the user depending on the tolerance to probable monogenic pathways. Finally, we chose to test the informativity of each pathway mode in decreasing order of variance explained in the dataset and to stop when a mode is found uninformative after mode 2 as we know the following will explain even less variance and is more likely to be noise.

**Predict cell type.** Once the activity matrix generated, a $k$-Nearest Neighbors matrix with $k = 20$ is computed, then an adjacency matrix using Jaccard distance and finally transformed as a weighted graph using igraph function graph.adjacency. Clustering is then performed using leiden_find_partition from leidenbase package with ModularityVertexPartition as partition type and a maximum number of iterations of 2. The average activity score is computed by cell type and by cluster. Each cluster is attributed the cell type for which the activity score is the highest, if it passes a threshold of default value 0, otherwise it is labeled as unassigned. This value can be modified by the user, depending on the level of confidence needed for annotation.

## Comparison with other tools

**Pagoda2 and AUCell to compare pathway activity scoring with MAYA.** Pagoda2 was run with default settings, following the vignette. AUCell was run using default settings, with log-normalized counts as input. Pagoda2 and AUCell were provided the same pathway lists as MAYA.

**Cell-ID and SCINA to compare cell type prediction with MAYA.** The two tools were provided the same PanglaoDB cell type marker lists as MAYA. Cell-ID was applied on a Seurat object following standard procedure, computing MCA and then performing hypergeometric test with gene lists. Each cell was attributed the cell type for which −log10(p-value) was the highest. When the value was inferior to 2, the cell was labeled unassigned. SCINA was run with default parameters, except for the sensitivity cutoff that was lowered to 0.9 and rm_overlap=FALSE as the input gene lists could be partially redundant.

**Integration with harmony.** Harmony was run through Seurat v3 with default settings.

**Integration using Canonical Correlation Analysis.** CCA was performed using Seurat and more specifically the functions FindIntegrationAnchors followed by IntegrateData. This function aims at identifying directions that account for the most co-variance between different datasets and is another popular method to perform batch effect correction.

**GSEA.** Fold change is computed using Seurat Function "FoldChange" with default parameters and comparing the population of interest with all other cells from the dataset. The package fgsea was used to compute the adjusted p-value of the enrichment of Hallmark pathway lists in differentially expressed genes, setting a threshold at 0.01 for significance.

**Non-negative Matrix Factorization.** We followed the method described by Gavish et al. to first identify the intra-tumoral heterogeneity programs in each patient individually using NMF on the subset of tumor cells from each patient, with different ranks. Only robust programs identified with different ranks were kept. We then merged into 'meta-programs' the robust individual programs that share the most similarity across patients based on their top50 genes, by taking the union of their contributing genes.

## Metrics

**Statistics.** A Wilcoxon rank sum test is systematically used to compare the distributions of two groups of data points and we provide a two-sided p-value to assess the significance of the test.

**Shannon Diversity Index.** It measures in each predefined cluster the diversity of cells in terms of patient identity, batch or cell type. Here we use it to measure the diversity of patients found in each Leiden cluster computed on the activity matrix.

$$SDI_c = \frac{(-1)^* \sum_{i=1}^{N} p_i^* \log(p_i)}{\log(N)} \tag{1}$$

With c the cluster in which we compute the SDI, N the number of different possible identities (patients in our case) and $p_i$ is the proportion of cells from the cluster corresponding to identity i. SDI of 1 indicates that cells constituting the cluster come equally from all possible identities i.e., the cluster displays high identity diversity.

**Specificity metric.** For a mode, we can compute for each predetermined cluster of cells (cells grouped by cell type in our case) a specificity score. As the sum of scores across clusters for a mode

equals 1, the maximum value of specificity across cells reflects the repartition of high activity scores between clusters.

$$S_{m,c} = \frac{a_{m,c}{}^2}{\sum_{p=1}^{N} a_{m,p}{}^2} \tag{2}$$

$$\sum_{p=1}^{N} S_{m,p}{}^2 = 1 \tag{3}$$

With $S_{m,c}$ the specificity of mode m in cluster c, $a_{m,c}$ the average activity score of m in c, and N the number of clusters.

We consider that specificity is significant for a cluster when it is 50% above expected value of 1/N (specificity score when all cells across all clusters have the same activity).

### Precision, recall, F1-score

$$Precision = \frac{TP}{TP+FP} \tag{4}$$

$$Recall = \frac{TP}{TP+FN} \tag{5}$$

$$F1\_score = \frac{2^* Precison^* Recall}{Precision + Recall} \tag{6}$$

Where *TP* is the number of true positives, *FP* the number of false positives and *FN* the number of false negatives. F1-score of 1 means perfect precision and recall.

### Matching PanglaoDB cell types with author annotation for precision and recall assessment

To assess precision and recall of cell-type annotation tools, we had to find equivalents of cell types described by authors in the PanglaoDB and chose the closest type or multiple types when PanglaoDB included several subtypes.

**Kidney.** Monocytes=c("Monocytes"), Endothelial cells=c("Endothelial cells"), Mesangial_cells=c("Mesangial cells","Smooth muscle cells"), Podocytes=c("Podocytes"), TCD8 = c("T cells","T memory cells", "T cytotoxic cells").

**Colon.** 'Mature Enterocytes'=c("Enterocytes"), 'Goblet cells'=c("Goblet cells"), Pericytes=c("Pericytes"), 'Smooth muscle cells'=c("Smooth muscle cells"), cDC=c("Dendritic cells"), Proliferating monocytes= c("Monocytes","Macrophages"), 'NK cells'=c("NK cells","Natural killer T cells"), 'Regulatory T cells'=c("T regulatory cells","T cells","T memory cells"), 'CD19 + CD20 + B' = c("B cells","B cells naive","B cells memory"), 'Mast cells'=c("Mast cells").

**Performances.** All tests were run with CPU: 6 cores/12 threads@ 2.6 GHz.

### Reporting summary
Further information on research design is available in the Nature Portfolio Reporting Summary linked to this article.

## Data availability
Kidney dataset: The count matrices were downloaded from Supplementary data S1 from Young et al. [https://www.science.org/doi/suppl/10.1126/science.aat1699/suppl_file/aat1699_datas1.gz.zip]. Colon dataset: Raw count matrix and cell annotations were downloaded from the NCBI Gene Expression Omnibus (GEO) database under the accession code GSE144735 for the KUL3 cohort. Ovary dataset: Count data were downloaded from the NCBI Gene Expression Omnibus (GEO) database

with accession code GSE165897. Larynx dataset: Count data were downloaded from the NCBI Gene Expression Omnibus (GEO) database with accession code GSE150321. Pancreas dataset: A Seurat object was directly loaded from the package SeuratData [https://github.com/satijalab/seurat-data] (panc8 v3.0.2). Breast dataset: Count matrix and metadata the dataset from Qian et al were retrieved from https://www.weizmann.ac.il/sites/3CA/breast as formatted by Gavish et al [https://www.dropbox.com/sh/nbx7v3om85wkfoq/AACpeZEZ4RNQwMW37Q7AHxExa?dl=1]. Lung dataset: Count matrix and metadata the dataset from Kim et al were retrieved from https://www.weizmann.ac.il/sites/3CA/lung as formatted by Gavish et al [https://www.dropbox.com/sh/byext689ffg77pj/AACp5jI2RRxndurKn2B0T-VWa?dl=1]. PBMC dataset: A Seurat object was directly loaded from the package SeuratData [https://github.com/satijalab/seurat-data] (pbmcsca v3.0.0). Reference databases: PanglaoDB was downloaded from the website (https://panglaodb.se/) [https://panglaodb.se/markers/PanglaoDB_markers_27_Mar_2020.tsv.gz]. MSigDB gene lists (Hallmark, KEGG and REACTOME) were downloaded from the Broad Institute website (http://www.gsea-msigdb.org/gsea/msigdb/collections.jsp) [https://data.broadinstitute.org/gsea-msigdb/msigdb/release/7.4/h.all.v7.4.symbols.gmt, https://data.broadinstitute.org/gsea-msigdb/msigdb/release/7.4/c2.cp.kegg.v7.4.entrez.gmt, https://data.broadinstitute.org/gsea-msigdb/msigdb/release/7.4/c2.cp.reactome.v7.4.symbols.gmt] in their version 7.4. Source data are provided with this paper.

## Code availability
MAYA is available as an R package on GitHub [https://github.com/One-Biosciences/MAYA/] and Zenodo [https://doi.org/10.5281/zenodo.7689013][66]. Requires $R \geq 4.0.5$. Code for reproducing data analysis and plots is available at [https://github.com/One-Biosciences/MAYA-figures/]. *Open Access* MAYA is available on GitHub at https://github.com/One-Biosciences/MAYA/ and licensed by One Biosciences under a GNU Affero General Public Licence v3.0. To view a copy of this license, visit https://www.gnu.org/licenses/agpl-3.0-standalone.html. Source data are provided with this paper.

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

## Acknowledgements

We thank members of the One Biosciences' team and the Vallot lab for testing MAYA on various devices and datasets and giving inputs on the manuscript.

## Author contributions

Y.L. and C.V., as scientific advisor for One Biosciences, conceived the algorithm. Y.L. implemented the code, C.V. supervised the work. Both authors wrote the manuscript.

## Competing interests

C.V. is a founder and equity holder of One Biosciences. The remaining author declares no competing interests.
