## [Peer Review File · Nature Communications]

Multi-modal quantification of pathway activity with MAYAREVIEWER COMMENTS

Reviewer #1 (Remarks to the Author):

In this study, the authors developed a computational method named MAYA to quantify the diverse modes of activation of biological pathways across cell populations, as well as predict cell types by using gene lists from reference markers. The MAYA also used the PCA method on a normalized gene-cell expression matrix restricted to pathway genes for detecting modes. The authors assumed that each principal component represents a mode of activation of the pathway, and defined the informativity of each mode based on the bimodal curve with multiple parameters. Compared to previous gene-set scoring methods, MAYA extended the application of multiple modes in pre-defined pathways on explaining the heterogeneity of tumor cells or immune cells. There were several major concerns influencing my confidence on this manuscript.

- 1) The authors used the PCA method to yield the principal components (PC) for defining different modes. What the proportion of the first PC explained variance of pathway? How many genes contribute to PC1 and PC2? Did this different gene numbers impact the results?
- 2) Seeing from the results, I noticed that the pathways mainly were grouped into two modes, few have four modes. Did the number of genes in pathways or biological relevance contribute to the number modes for each pathway?
- 3) As we known, there exist high inter-connected among pathways. Did these similar pathways influence the results of the MAYA?
- 4) For the function of automatically annotating cell identity, the performance of MAYA is largely influenced by the accuracy of reference marker lists (e.g. PanglaoDB). Some known cell types have only very few markers, as well as some cell types have largely similar cell markers. The authors should give more evidence for the performance of method on such cases.

Reviewer #2 (Remarks to the Author):

Landais and Vallot have developed what seems to be an important tool that fills a critical gap in our armoury of analytical tools, namely a method for quantifying pathway activity in single cell mRNA data. I think the authors provide sufficient evidence to demonstrate that their method is sound and produces plausible results. Conceptually I find it very convincing, and I was particularly impressed that the method manages to unite cancer cells from different individuals without regressing out biological differences. Whether or not MAYA will become the dominant tool for the annotation of pathway activity, will become apparent in time. At the very least MAYA is a serious contender. I do not have any major criticism. The paper is clear and well written. One thing that the authors may want to consider is to be less critical of other methods in their discussion. There is a way of stating the superiority of one's method without explicitly saying so.

Sam Behjati

Reviewer #3 (Remarks to the Author):

The authors present a new bioinformatics tool, MAYA, for pathway analysis in single-cell transcriptomics data. Instead of considering pathway gene lists as a whole, MAYA detects subgroups of genes that are coordinately expressed within each pathway gene list, in a process called "multi-modal" pathway analysis. The authors confirm that reference pathway signatures, eg in KEGG and REACTOME, can be heterogeneous, where different sets of genes within the pathway are expressed by different cell populations. This is indeed a potentially critical limitation of broadly used pathway analysis applied to scRNA-seq data that deserves more research. Overall we find the idea behind MAYA interesting, and the tool might be appealing to researchers conducting scRNA-seq data analysis. We also believe that the biological significance and relevance of results generated by the tool are not currently well substantiated, and present several concerns.

Major comments:

1- The whole point of doing pathway analysis is to provide a straightforward interpretation of biological processes. If a pathway signature is too ambiguous or of low-quality, decomposing it in 'modes' is not likely to solve the problem.

How is the researcher supposed to interpret the functional relevance of each mode?

If pathway A is enriched in population A and not B; but pathway A mode 2 is expressed in population B but not A; what's the interpretation?

If pathway signatures are not trustworthy, wouldn't it be better not to use them, and instead do directly differential gene expression analysis and manual interpretation?

2- Relatedly, use of modes of activation within pathway analysis framework would require validating them across a large number of datasets. Eg did the authors confirm that the found "modes of activation of TNF pathway" are reproducible in multiple datasets? How do these different modes of activation differ biologically and how to interpret them?

3- In cases shown, gene subsets found don't seem to be 'modes of activation' of a pathway but rather sub-type markers deconvolution from a bulk-generated signature. As an example, the Allograft rejection pathway presents two modes of activation: mode 1, driven by the expression of CTSS and SPI1 (monocyte markers), and mode 2, driven by CD2, CD3E, CD3D, CD8A, CD8B (CD8 T cell markers). These genes are always mutually exclusive markers of different immune cell populations, only confirming that the pathway is too broad (tissue-derived) to analyze single-cell data. My interpretation from this is that an expert analyst should not use this signature which is not appropriate for single-cell analysis. Perhaps, it would be useful for the broad community if the authors would comprehensively process the pathway collections, and flag those like this one that are not useful for single-cell analysis as this one.

4- About the algorithm:

a) rows of the matrix are scaled so that more highly expressed genes do not weight more than the others in the PCA that is later performed.

Isn't there a threshold for minimal variance? Re-scaling of very lowly expressed genes would strongly amplify noise. Is this not an issue for MAYA?

Can you show this is not the case in both simulations and eg MSigDB gene sets?

b) "we interrogate PC2 even if PC1 is uninformative, as PC2 can still explain a significant part of the variance" Can you show evidence for this? How much variance is explained by PC2 compared with eg PC3 on average across signatures?

c) "The Informativity metric depends on finding bimodal distributions."

what if a signature is homogeneously high? for instance eg mode 1 of TNFA evaluated on a dataset of monocytes only? or Allograft rejection Mode 2 in T cells? MAYA would not be able to detect the signal, making the output dependent on the composition of the dataset, which is not desirable.

The authors could evaluate this and discuss this potential limitation.

Also, how robust is this metric to different technologies that produce different gene expression distributions (eg smart-seq2/3 vs 10x 3'/5') ?

This metric seems to have a major impact on the output of MAYA, and depends on how the density curve is adjusted.

Have the authors systematically evaluated the robustness of their method on 'positive control' modes with different numbers of genes and different gene mean expression values? (eg obtained by differential expression analysis)

5- Being a bioinformatics resource, it would be critical to include documented code to reproduce the main analysis and especially the comparison with the other methods.

6- The authors show that MAYA scores lead to better cell type prediction than AUCell, Pagoda2, etc.. However those methods were not designed for cell type annotation; how MAYA would compare with more relevant signature-based classification methods such as scGate, SCINA or Garnett?

Also, it is not clear if in this context MAYA is benefiting from 'multi-modal' pathway as well ?

MAYA's function as an automated cell type framework is overstated, because highly specific signatures are required as input, which are not provided by MAYA. In fact, in the example provided MAYA fails to discriminate between NK and T cells.

7- In the batch effect analysis, is the patient effect also present if using the intersection of variable genes across samples ? (this is standard practice, eg using Seurat FindIntegrationAnchors)
"MAYA had an average SDI of 0.77 against 0.65 and 0.17 for the integration-based and the gene-based method respectively (Fig 4h)" Is this significant and reproducible in other datasets?

8- In general we find the claims about the 'discovery' capacity overstated, as many different results can be presented as self-confirmatory.
eg "we show that MAYA identifies several modes of pathway activation shared across patients (Fig. 5a,b and Supplementary Fig. 5a,b) that are associated with known cancer hallmarks."
However no evidence is shown that MAYA is better than standard approaches, eg using GSEA; or that the 'modes' are not highly correlated with whole signature scores (eg KRAS_SIGNALING_DN, ESTROGEN_RESPONSE_EARLY, etc.)
Are all these tumors driven by KRAS or estrogen response?
In fact cancer cells show enrichment in KRAS_SIGNALING_DN (DN="down-regulated" genes) while macrophages are enriched in KRAS_SIGNALING_UP.
Do the authors suggest that cancer cells down-regulate KRAS signaling while macrophages up-regulate this pathway?

Also "It notably detects a cell-type specific activation of complement genes in macrophages." Why is this notable? Is this proven or have a demonstrated role in these tumors?

"MAYA identifies common modes of activation across tumor cells, which could be compared to such tumor meta-programs" But no comparison is provided; so how to interpret these "modes of activation"?

9- "MAYA focuses on cell identity by looking only at genes considered as markers, it does not detect the variations between patients driven by other sets of genes that are not related to cell type identity and that lead to the formation of different clusters in a classical gene-based analysis." A fair comparison would be to do clustering using the same set of cell type discriminating genes. The differences shown might be not due to MAYA, but to the list of genes provided to it as input.

Minor comments

- In the Kidney dataset, when matching labels for PanglaoDB "T helper cells", those are CD4 and not CD8 T cells. Same for "Regulatory T cells" for the colon dataset. All the other categories which are merged are not regulatory T cells.
- we find confusing the use of "oncogenic" to describe tumor/cancer datasets
- the use of "multi-modal" can be misleading as this term is broadly used in the single-cell biology field for multi-omics assays. Consider alternatives, eg multi-modular
- line 88: However, not all detected modes might not reflect a relevant biological pattern in the

RESPONSE TO REVIEWERS' COMMENTS

Reviewer #1 (Remarks to the Author):

In this study, the authors developed a computational method named MAYA to quantify the diverse modes of activation of biological pathways across cell populations, as well as predict cell types by using gene lists from reference markers. The MAYA also used the PCA method on a normalized gene-cell expression matrix restricted to pathway genes for detecting modes. The authors assumed that each principal component represents a mode of activation of the pathway, and defined the informativity of each mode based on the bimodal curve with multiple parameters. Compared to previous gene-set scoring methods, MAYA extended the application of multiple modes in pre-defined pathways on explaining the heterogeneity of tumor cells or immune cells. There were several major concerns influencing my confidence on this manuscript.

1) The authors used the PCA method to yield the principal components (PC) for defining different modes. What the proportion of the first PC explained variance of pathway?

Following the reviewer's comment, we have now computed the variance explained by the first five principal components of the PCA (Fig. 1R1), when running MAYA with various reference databases (Hallmark, KEGG or REACTOME) on two datasets (kidney and colon datasets). In average PC1 explains respectively 7.1%, 10% and 8.6% of the variance of each pathway, showing that one component is not sufficient to capture the biological complexity of the pathways. Several modes/PCs are needed to account for the biological variation within each pathway. We have now included Fig. 1R1 in the manuscript (Supplementary Fig. 1).

Figure 1R1: Overlaid jitter and boxplot representations of the fraction of variance explained by PC1 to PC5, for a PCA computed with either (a) Hallmark pathways on the normal kidney dataset, and (b) KEGG or (c) REACTOME pathways on the normal colon dataset.

How many genes contribute to PC1 and PC2? Did this different gene numbers impact the results?

There are on average more genes contributing to PC1 than to PC2 when running MAYA with various reference databases (Hallmark, KEGG or REACTOME) on two datasets (normal kidney and colon datasets, Fig. 1R2). Such difference might reflect the lower variance explained by PC2 in comparison

to PC1 (see above Fig. 1R1). This observation is inherent to the usage of the PCA algorithm, and does not seem to influence MAYA, as it successfully detects two modes for the majority of pathways.

Figure 1R2: Boxplot representation of the number of contributing genes for PC1 and PC2 for Hallmark pathways on the kidney dataset, and KEGG or REACTOME pathways on the colon dataset. Contributing genes have a contribution superior to 0.1 for the corresponding PC.

2) Seeing from the results, I noticed that the pathways mainly were grouped into two modes, few have four modes. Did the number of genes in pathways or biological relevance contribute to the number modes for each pathway?

We agree with the reviewer that the size of input gene lists may influence the number of modes that MAYA detects. To test this, we generated simulated gene lists of different size. We used top markers ($n=2$ to 20 markers) from one, two or three populations within the normal colon dataset; these top markers were then diluted within random genes to make simulated gene lists of various sizes ($n=10$ to 60 genes in total). In theory, MAYA should detect respectively one, two or three activation modes from these simulated gene lists ($n=1,400$ gene lists). We measured the ability of MAYA to detect the expected number of modes using a F1-score (Fig. 1R3).

Overall, whatever the number of expected modes, we show that the number of contributing genes is determinant for MAYA's performance, independently of the total gene list size. MAYA detects one, two or three modes starting from $n=10$ contributing genes, whatever the size of the total gene list with a minimum of $n=20$ (Fig. 1R3a). It confirms that MAYA will not detect modes that are driven by very few genes and confirms the default choice in the MAYA algorithm to consider only gene lists with more than 10 genes. Most gene lists within the studied database are above this threshold (Fig. 1R3b, only 27/178 PanglaoDB gene lists with less than 10 genes, no gene list below 10 genes in other databases).

Regarding the effect of biological relevance on mode detection, we had already addressed this point by testing MAYA's ability to properly detect relevant modes of activation starting from noisy gene lists (see Fig. 2f and Supplementary Fig. 2e).

Figure 1R3: (a) Overlaid jitter and boxplot representation of F1-scores for the detection of the expected number of simulated modes, according to the number of top markers used to make simulated gene list and the size of the final simulated gene list and the number of expected modes. (b) Overlaid jitter and boxplot representation of the size of the gene lists from the databases MSigDB Hallmark, KEGG, REACTOME and PanglaoDB.

3) As we known, there exist high inter-connected among pathways. Did these similar pathways influence the results of the MAYA?

We agree with the reviewer that gene lists from databases could be redundant. Yet, MAYA inspects each gene list independently, in a sequential fashion by a dedicated PCA for each gene list. Therefore, the results for one gene list, aka the identified modes, cannot influence the results for another gene list. It is up to the user to interpret and compare modes of activation between pathways/gene lists.

Nevertheless, we have checked if pathways from different databases had a substantial overlap in terms of genes (Fig. 1R4). We noticed that the overlap was negligible for the Hallmark and KEGG databases but was stronger for the REACTOME database, probably resulting in more redundancy between the identified modes. In the precise case of cell type annotation, we had already implemented a function to resolve gene list overlap and propose a single annotation for each cell, only selecting the mode with the highest activation score.

Figure 1R4: Heatmap representation of the number of shared genes across (a) Hallmark, (b) KEGG and (c) REACTOME databases.

4) For the function of automatically annotating cell identity, the performance of MAYA is largely influenced by the accuracy of reference marker lists (e.g. PanglaoDB). Some known cell types have only very few markers, as well as some cell types have largely similar cell markers. The authors should give more evidence for the performance of method on such cases.

We agree with the reviewer that MAYA's performance depends on the quality of the reference marker lists, but just like any other tool for automatic cell annotation. To further test the performance of MAYA in complex cases and compare it to other tools (now SCINA in addition to Cell-ID), we first simulated marker lists of various sizes from PanglaoDB, by randomly selecting several times a predetermined number of markers by cell type, ranging from 10 to 50. We then provided these gene lists as input to MAYA and to two other cell type prediction tools, Cell-ID and SCINA, and assessed their accuracy for cell type prediction by computing a F1-score by cell type (Fig. 1R5). Altogether, MAYA – as shown in Fig. 1R3 – needs a minimum of 20 genes per list to assign cell identity, and above this threshold it regularly outperforms existing tools- although results become more robust with 40-50 genes.

Figure 1R5: Overlaid jitter and boxplot representations of the F1-score for each cell type and each prediction tool, according to the maximum number of selected genes in PanglaoDB markers lists (a two-sided Wilcoxon test is used to compare MAYA with the two other methods, adjusted p-values are symbolized with: *: <0.05, **: <0.01, ***: <0.001, ****: <0.0001).

Regarding the redundancy of marker lists in databases, this is indeed something that we noticed in the PanglaoDB database, where some types of immune cells share more than 60% of their markers (Fig. 1R6a). As mentioned above (Point 3/), to circumvent this issue, we had initially added a function in MAYA to only retain the cell type annotation that had the highest average activity in the cluster as final annotation (example in Fig. 1R6b-c). In our view, this selects the most relevant annotation. However, the user can always come back to the matrix of activity by cell type and cluster to check if other annotations could have been possible for the cluster.

Figure 1R6: (a) Barplot displaying the percentage of overlap between PanglaoDB cell type marker lists used as input for MAYA to annotate the kidney dataset, ordered by decreasing overlap and only showing the top 20 pairs with the highest overlap. (b) Heatmap representation of the average activity score by PanglaoDB cell type and MAYA cluster. (c) UMAP representation of activity matrix of PanglaoDB marker lists, cells are colored according to MAYA clusters and MAYA annotation.

Reviewer #2 (Remarks to the Author):

Landais and Vallot have developed what seems to be an important tool that fills a critical gap in our armoury of analytical tools, namely a method for quantifying pathway activity in single cell mRNA data. I think the authors provide sufficient evidence to demonstrate that their method is sound and produces plausible results. Conceptually I find it very convincing, and I was particularly impressed that the method manages to unite cancer cells from different individuals without regressing out biological differences. Whether or not MAYA will become the dominant tool for the annotation of pathway activity, will become apparent in time. At the very least MAYA is a serious contender. I do not have any major criticism. The paper is clear and well written. One thing that the authors may want to consider is to be less critical of other methods in their discussion. There is a way of stating the superiority of one's method without explicitly saying so.

Sam Behjati

We thank the reviewer for acknowledging the interest and capacities of MAYA. Following the reviewer's recommendation, we have now removed from the second paragraph of the discussion (lines 294-300) the part where we described the limitations of Pagoda2 and AUCell, to solely focus on the specificities of MAYA.

Reviewer #3 (Remarks to the Author):

The authors present a new bioinformatics tool, MAYA, for pathway analysis in single-cell transcriptomics data. Instead of considering pathway gene lists as a whole, MAYA detects subgroups of genes that are coordinately expressed within each pathway gene list, in a process called "multi-modal" pathway analysis. The authors confirm that reference pathway signatures, eg in KEGG and REACTOME, can be heterogeneous, where different sets of genes within the pathway are expressed by different cell populations. This is indeed a potentially critical limitation of broadly used pathway analysis applied to scRNA-seq data that deserves more research. Overall we find the idea behind MAYA interesting, and the tool might be appealing to researchers conducting scRNA-seq data analysis. We also believe that the biological significance and relevance of results generated by the tool are not currently well substantiated, and present several concerns.

Major comments:

1) The whole point of doing pathway analysis is to provide a straightforward interpretation of biological processes. If a pathway signature is too ambiguous or of low-quality, decomposing it in 'modes' is not likely to solve the problem. How is the researcher supposed to interpret the functional relevance of each mode? If pathway A is enriched in population A and not B; but pathway A mode 2 is expressed in population B but not A; what's the interpretation?

We agree with the reviewer that the quality of pathway analysis greatly depends on the quality of the input list, and this holds true whatever the tool that is used. Using MAYA on such lists will indeed not

solve the issue and we believe it is the responsibility of the user to be aware of the input lists he uses for pathway analysis. We have however demonstrated in this manuscript that the most widely used gene lists (MSigDB Hallmark, KEGG and REACTOME) could be used confidently as input for MAYA and provide relevant results (see below for additional example). In addition, we have shown in our signal dilution experiment (Fig. 2f and Supplementary Fig. 2e) that MAYA activity scores were stable when adding random genes to a gene list, which is one case of low-quality gene lists currently encountered by biologists.

Regarding the question of the functional relevance, detecting modes of activation specific to cell populations is the purpose of MAYA (Pathway A mode 1 in population X and not Y, and A mode 2 in population Y and not X). The interpretation is that population X uses a different set of genes to activate the pathway A than population Y. In addition to the examples of Epithelial Mesenchymal Transition, Estrogen response early and Coagulation presented in the original version of the manuscript, we have added the example below (Fig. 3R1). We study the KEGG Wnt pathway, where we detect 2 modes of activation, one specific to tumor cells and the other specific to Cancer Associated Fibroblasts (CAF). Inspecting the top contributing genes, we conclude that mode 1 corresponds to the canonical Wnt pathway while mode 2 corresponds to the noncanonical Wnt pathway. Yet another example of how MAYA untangles relevant modes of pathway activation across cell populations. We have added this example to further support the utility of MAYA for pathway analysis in the manuscript lines 262-267, Fig. 5.

Figure 3R1: (a) Heatmap of activity scores of the two modes of the KEGG Wnt pathway, initial author annotation is indicated above heatmap. (b) Heatmap of scaled gene expression for top10 contributing genes for the two modes of Wnt pathway, ordered by decreasing contribution. (c) Violin plots of activity scores for the two modes of Wnt pathway, grouped by author annotation (adjusted p-values from Wilcoxon test are symbolized with: *: <0.05, **: <0.01, ***: <0.001, ****: <0.0001).

If pathway signatures are not trustworthy, wouldn't it be better not to use them, and instead do directly differential gene expression analysis and manual interpretation?

Following the reviewer's concern, we have now compared MAYA to standard approaches, as GSEA (Fig.3R2) post differential analysis. We show that MAYA enables the detection of a variety of modes of activation, that GSEA did not pick up (panel a vs b of Fig. 3R2). We have added this comparison in the manuscript lines 234-235, Supplementary Fig. 5.

Figure 3R2: (a) Barplot displaying Hallmark pathways significantly enriched using GSEA in over-expressed genes in tumor cells, macrophages and CAF. x-axis corresponds to $-\log_{10}$ adjusted p-values. (b) Barplot representation of specificity scores of the top specific modes for tumor cells, macrophages, and CAF. In bold are indicated the gene lists found with GSEA analysis.

In particular, an interesting case is the EMT pathway: MAYA detects a tumor-specific mode of EMT while GSEA results suggest that the EMT genes are significantly under-expressed in these cells (Fig. 3R3a-b). The negative fold changes of genes overexpressed in CAF and macrophages actually hide the over-expression of other EMT genes specific to tumor cells (Fig. 3R3c).

Figure 3R3: (a) Barplot displaying Hallmark pathways significantly enriched using GSEA in under-expressed genes in tumor cells. (b) Heatmap of activity scores of the three modes of EMT pathway, initial author annotation is indicated above heatmap. (c) Overlaid jitter and boxplot representation displaying the log Fold Change of top30 contributing genes to each EMT mode, when comparing tumor cells to all other cells (adjusted p-values from Wilcoxon test are symbolized with: *: <0.05, **: <0.01, ***: <0.001, ****: <0.0001).

2) Relatedly, use of modes of activation within pathway analysis framework would require validating them across a large number of datasets. Eg did the authors confirm that the found “modes of activation of TNF pathway” are reproducible in multiple datasets? How do these different modes of activation differ biologically and how to interpret them?

To show the reproducibility of MAYA modes across datasets, we ran MAYA on two additional cancer datasets from the breast and the lung using the Hallmark database. Focusing on the EMT and Coagulation pathways that we presented in the original version of the manuscript (Fig. 5 and Supplementary Fig. 5), we show that the CAF and tumor cell specific modes of EMT are also found in the breast and lung datasets, with a set of similar top contributing genes. Similarly for the Coagulation pathway, the CAF and macrophages specific modes are both reproducible across datasets. We have now added this analysis in the manuscript lines 253-255, Supplementary Fig. 5.

Figure 3R4: (a) Heatmap of activity scores of the different modes of EMT pathway in the ovary, breast and lung datasets, consensus annotation is indicated above heatmap. (b) Heatmap displaying the presence of genes in the top20 contributing genes of the EMT mode specific to CAF (respectively tumor cells) of the three datasets. (c) Heatmap of activity scores of the different modes of Coagulation pathway in the ovary, breast and lung datasets, consensus annotation is indicated above heatmap. (d) Heatmap displaying the presence of genes in the top20 contributing genes of the Coagulation mode specific to CAF (respectively Macrophages) of the three datasets.

3) In cases shown, gene subsets found don't seem to be 'modes of activation' of a pathway but rather sub-type markers deconvolution from a bulk-generated signature. As an example, the Allograft rejection pathway presents two modes of activation: mode 1, driven by the expression of CTSS and SPI1 (monocyte markers), and mode 2, driven by CD2, CD3E, CD3D, CD8A, CD8B (CD8 T cell markers). This genes are always mutually exclusive markers of different immune cell populations, only confirming that the pathway is too broad (tissue-derived) to analyze single-cell data. My interpretation from this is that an expert analyst should not use this signature which is not appropriate for single-cell analysis. Perhaps, it would be useful for the broad community if the authors would comprehensively process the pathway collections, and flag those like this one that are not useful for single-cell analysis as this one.

As mentioned above, we agree with the reviewer that gene lists can be of varying quality. We believe it is out of the scope of MAYA to curate gene lists and flag them, which is the full-time job of the entire community. Yet to further demonstrate the ability of MAYA to detect relevant modes of activation that are not only directly related to cell type, we show here the example of the Estrogen response early pathway in the breast cancer dataset for which MAYA detects multiple modes of activation within tumor cells, corresponding to different patient phenotypes, and not to cell identity.

Figure 3R5: (a) Heatmap of scaled gene expression for top10 contributing genes for the four modes of Estrogen response early pathway specific to tumor cells in the breast dataset, ordered by decreasing contribution. (b) Heatmap of activity scores in tumor cells of the four modes of Estrogen response early pathway specific to tumor cells in the breast dataset, sample annotation is indicated above heatmap. (c) Violin plots of activity scores in tumor cells for the four modes of Estrogen response early pathway specific to tumor cells in the breast dataset, grouped by sample (adjusted *p*-values from Wilcoxon test are symbolized with: *: <0.05, **: <0.01, ***: <0.001, ****: <0.0001).

4) About the algorithm:

a) rows of the matrix are scaled so that more highly expressed genes do not weight more than the others in the PCA that is later performed.

Isn't there a threshold for minimal variance?

We did not set a threshold for minimal fraction of variance explained by a PC as we could not find an empirical cutoff that would apply to all datasets and databases rationally. Looking at Fig. 3R6, we see that fraction of explained variance varies across databases and datasets. We have now included Fig. 3R6 in the manuscript (Supplementary Fig. 1) to further support this choice.

Figure 3R6: Overlaid jitter and boxplot representations of the variance explained by PC1 to PC5, for a PCA computed with either (a) Hallmark pathways on the normal kidney dataset, and (b) KEGG or (c) REACTOME pathways on the normal colon dataset.

Re-scaling of very lowly expressed genes would strongly amplify noise. Is this not an issue for MAYA? Can you show this is not the case in both simulations and eg MSigDB gene sets?

Thanks to the reviewer’s comment, we have now compared MAYA results with or without gene scaling prior to PCA computing (Fig. 3R7). We have done so using the three usual pathway databases on the normal kidney and colon datasets. We found that not scaling genes increased the number of monogenic modes especially for the KEGG and REACTOME databases, but that these monogenic modes were mostly filtered out when applying default MAYA cutoff on maximum gene contribution to a mode. Additionally, we did not detect a significant difference in the specificity of selected modes. Given these results, we decided to keep the default setting in MAYA to a ‘scaling’ mode while enabling the user to change it to ‘unscaling’ if wanted.

Figure 3R7: (a) Scatter plot representation of the maximum gene contribution versus the number of cells above bimodality threshold for each mode detected as bimodal by MAYA with different datasets and input gene lists. MAYA default thresholds for these two parameters are indicated with a dashed line (maximum contribution of 0.4 and number of cells above threshold superior to 5% of the total number of cells). When computing PCA, gene expression was either centered and scaled (default MAYA settings), or only centered. Monogenic modes, defined as modes with a contribution of the second most contributing gene is less than 30% of the contribution of the most contributing gene, are colored in black. The proportions of monogenic and multigenic modes before and after applying the filter on maximum gene contribution are displayed as barplots under each corresponding scatter plot. The filter on the number of cells above threshold was applied for both. (b) Boxplot representation of the specificity of modes kept after filtering on maximum gene contribution and minimum number of cells above threshold, with or without scaling of gene expression before PCA computation in MAYA algorithm, for different datasets and input gene lists.

b) “we interrogate PC2 even if PC1 is uninformative, as PC2 can still explain a significant part of the variance” Can you show evidence for this? How much variance is explained by PC2 compared with eg PC3 on average across signatures?

We have now computed the explained variance for several PCs and across databases and datasets (Fig. 3R8). We agree with the reviewer that the difference in the variance explained on average by PC1 and PC2, and PC2 and PC3 does not alone explain why we chose to consider a PC when the previous one is

uninformative. This choice was rather empirical, based on the observation that these additional selected modes – when searching beyond PC2 when PC1 & 2 are uninformative - had a significantly lower specificity, especially for the KEGG and REACTOME databases (Fig. 3R8d).

Figure 3R8: (a-c) Overlaid jitter and boxplot representations of the variance explained by PC1 to PC5, for a PCA computed with either (a) Hallmark pathways on the kidney dataset, and (b) KEGG or (c) REACTOME pathways on the colon dataset. (d) Overlaid jitter and boxplot representation displaying the specificity of modes detected in MAYA default setting and the specificity of additional modes only selected when removing the constraint that all previous modes should be informative (adjusted p-values from Wilcoxon test are symbolized with: * : <math>< 0.05</math>, ** : <math>< 0.01</math>, *** : <math>< 0.001</math>, **** : <math>< 0.0001</math>).

c) “The Informativity metric depends on finding bimodal distributions.” what if a signature is homogeneously high? for instance eg mode 1 of TNFA evaluated on a dataset of monocytes only? or Allograft rejection Mode 2 in T cells? MAYA would not be able to detect the signal, making the output dependent on the composition of the dataset, which is not desirable. The authors could evaluate this and discuss this potential limitation.

MAYA does indeed require heterogeneity to detect modes of activation, just like a standard differential analysis would. We do not find it a major limitation to the tool as single-cell datasets often contain more than one cell type or phenotype that introduce heterogeneity in the data which allows the use of MAYA, on the full dataset to detect homogeneous activation in a cell type or on a subpopulation directly if relevant (comparing control T cells with T cells stimulated with interferon beta for instance). However, we included a discussion of this matter in the discussion lines 298-300 to further underline the importance of heterogeneity in the dataset in order to use the full potential of MAYA.

Also, how robust is this metric to different technologies that produce different gene expression distributions (eg smart-seq2/3 vs 10x 3'/5')?

As pointed out by the reviewer, all four example datasets were generated using 10X Chromium technology, either v2 or v3. To show that MAYA, and more specifically its informativity metric, do not depend on the sequencing technology that is used, we ran MAYA with Hallmark pathways on two PBMC datasets, one generated using 10X Chromium v3 (10X) and the other Smart-Seq2 (SS2). The two datasets have approximately the same cell populations represented in similar proportions (Fig. 3R9). The main differences between the datasets are the number of cells (2,993 for 10X and 526 for SS2 after

QC), and the coverage (median gene coverage of 1,512 and 2,508 for 10X and SS2 respectively). We found that the overlap between the pathways detected with at least one informative mode was significant and so was the number of modes detected in both pathways for the pathways in common (Fig. 3R9c). Furthermore, these common modes across datasets shared on average 7 of their top10 contributing genes.

In addition, our informativity method is not directly applied to the distribution of the expression of a single gene across cells, but to the distribution of the coordinates of cells on a given principal component. We show with three pathway modes that the bimodality detection performs successfully for both single-cell technologies (Fig. 3R9e). We have included this analysis in the manuscript lines 102-104, Supplementary Fig. 1.

Figure 3R9: (a) Violin plots displaying the gene coverage and percentage of mitochondrial RNA by cell in the 10X Chromium and Smart-Seq2 datasets. (b) Barplot representations of the cell type proportions in the 10X Chromium and Smart-Seq2 datasets. (c) Venn diagram displaying respectively the intersection of pathways selected by MAYA and the intersection of modes for common pathways in the 10X Chromium and Smart-Seq2 datasets. Significance of the overlap is assessed using Fisher's exact test. (d) Barplot representation of the number of genes overlapping between top10 contributing genes for modes selected in both 10X Chromium and Smart-Seq2 datasets. (e) Examples of distributions of activity scores for three modes in 10X Chromium and Smart-Seq2 datasets. Bimodality thresholds are marked by a vertical dashed line.

This metric seems to have a major impact on the output of MAYA, and depends on how the density curve is adjusted. Have the authors systematically evaluated the robustness of their method on 'positive control' modes with different numbers of genes and different gene mean expression values? (eg obtained by differential expression analysis)

Thanks to the reviewer comment, we have now simulated gene lists to test the effect of the total gene list size, the number of contributing genes (Fig. 3R10a) and the effect of gene expression levels on mode detection (Fig. 3R10b).

We used top markers (n=2 to 20 markers) from one, two or three populations within the normal colon dataset; these top markers were then diluted within random genes to make simulated gene lists of various sizes (n=10 to 60 genes in total). In theory, MAYA should detect respectively one, two or three activation modes from these simulated gene lists (n=1,400 gene lists). We measured the ability of MAYA to detect the expected number of modes using a F1-score (Fig. 3R10a). For a second experiment (n=180, Fig. 3R10b), we fixed the number of contributing genes to 15 and the final list size to 60, and randomly picked contributing genes of low, medium or high expression.

Overall, simulations show that MAYA accurately detects the expected number of modes for lists that contain modes driven by more than 10 genes, whatever the size of the final list. In addition, MAYA seems quite robust to the expression levels of contributing genes, even though accuracy is higher for highly expressed genes.

Figure 3R10: (a) Overlaid jitter and boxplot representation of F1-scores for detection of the expected number of simulated modes by MAYA, according to the number of top markers used by simulated mode, the size of the final simulated gene list and the number of expected modes. (b) Overlaid jitter and boxplot representation of F1-scores for detection of the expected number of simulated modes by MAYA, according to the expression level of the top markers used to simulate modes and the number of expected modes.

5) Being a bioinformatics resource, it would be critical to include documented code to reproduce the main analysis and especially the comparison with the other methods.

In the initial version of the manuscript, we had only provided the code for MAYA R package. We have now included the code to reproduce the main figures and analyses of the manuscript in our Github repository, as mentioned in Code availability section of the revised manuscript (line 485).

6) The authors show that MAYA scores lead to better cell type prediction than AUCell, Pagoda2, etc.. However those methods were not designed for cell type annotation; how MAYA would compare with more relevant signature-based classification methods such as scGate, SCINA or Garnett?

We agree with the reviewer that AUCell and Pagoda2 were not designed for cell type annotation. We had added them initially because they output – like MAYA - an activity matrix on which clustering can be performed and clusters can be attributed the identity that they activate the most. Cell-ID, on the other hand, was designed for cell type annotation. Following the reviewer’s suggestion, we have now added a second signature-based classification tool in addition to Cell-ID, SCINA, a tool that can take broad gene lists as input of the function and that does not require prior clustering (Fig. 3R11). We see that it provides results that are quite similar to MAYA in terms of precision and recall, with no statistical differences between the distribution of their F1-scores for the colon dataset but that it presents lower

recall on the kidney dataset. We have now integrated these plots focusing on cell annotation tools solely in Fig. 4 and Supplementary Fig. 4 of the revised version of the manuscript.

Figure 3R11: (a,c) Overlaid jitter and boxplot representation of (a) F1-scores, (b) precision and (c) recall for automatic annotation of the kidney dataset using Cell-ID, SCINA and MAYA, datapoints are colored according to author annotation. (d,f) Overlaid jitter and boxplot representation of (d) F1-scores, (e) precision and (f) recall for automatic annotation of the colon dataset using Cell-ID, SCINA and MAYA, datapoints are colored according to author annotation.

Also, it is not clear if in this context MAYA is benefiting from 'multi-modal' pathway as well?

The MAYA function for automated cell type annotation does not benefit from the multimodality of the analysis. For each gene list, we only look at PC1. If the associated activity score is informative, it is integrated to the final activity score matrix that is then used to cluster cells. We have now clarified this in the text line 307.

MAYA's function as an automated cell type framework is overstated, because highly specific signatures are required as input, which are not provided by MAYA. In fact, in the example provided MAYA fails to discriminate between NK and T cells.

The claim of “automated” cell type annotation has been modified to “assisted” cell type annotation, as we agree with the reviewer that the annotation provided by MAYA does need human intelligence afterwards to refine potential annotations. We only claim that it does not require prior clustering and want to show that it provides competitive results with other tools specialized in cell type annotation. We have now added a discussion of these limitations lines 310-313.

7) In the batch effect analysis, is the patient effect also present if using the intersection of variable genes across samples ? (this is standard practice, eg using Seurat FindIntegrationAnchors). “MAYA had an average SDI of 0.77 against 0.65 and 0.17 for the integration-based and the gene-based method respectively (Fig 4h)” Is this significant and reproducible in other datasets?

As suggested by the reviewer, we have now added an additional batch effect correction method to our benchmark on the larynx dataset presented in the manuscript (Fig. 4h), based on ‘Seurat’ Canonical Correlation Analysis, that aims at finding directions that account for the most co-variance between

different datasets. MAYA does significantly achieve better diversity of patients in its clusters than the gene-based method but the difference with the other batch correction methods is not significant.

To assess the reproducibility of MAYA to correct batch effect, we tested MAYA on an additional dataset of pancreas samples generated with 5 different technologies - CEL-Seq (1 & 2), Fluidigm C1, Smart-Seq2 and inDrop - for a total of 14,890 cells (Fig. 3R12b). MAYA, as Harmony and Seurat CCA, properly integrates all technologies. The analysis on the pancreas dataset was included in the manuscript lines 213-217, Supplementary Fig. 4.

Figure 3R12: (a,b) Overlaid jitter and boxplot representation of Shannon Diversity Index (SDI), for clusters derived from gene-based dimensionality reduction, Harmony dimensionality reduction, SeuratCCA and MAYA activity matrix of the (a) larynx and (b) pancreas datasets.

8- In general we find the claims about the 'discovery' capacity overstated, as many different results can be presented as self-confirmatory. eg "we show that MAYA identifies several modes of pathway activation shared across patients (Fig. 5a,b and Supplementary Fig. 5a,b) that are associated with known cancer hallmarks." However no evidence is shown that MAYA is better than standard approaches, eg using GSEA; or that the 'modes' are not highly correlated with whole signature scores (eg KRAS_SIGNALING_DN, ESTROGEN_RESPONSE_EARLY, etc.)

Following the reviewer's concern, we have now compared MAYA to standard approaches, as GSEA, as shown in Figure 3R2-3. We show that MAYA enables the detection of additional cell type specific modes of activation, that the GSEA analysis did not pick up. This comparison was included in Supplementary Fig.5 of the manuscript.

In addition, following the reviewer's recommendations, we checked whether the scores of the different MAYA modes were correlated with the whole signature score – i.e average expression of the genes of the corresponding pathway in each cell (Fig. 3R13). If the first MAYA mode is highly correlated to average pathway expression, we show that other modes are barely correlated to the average expression score of the pathway. These results precisely highlight the specificity of MAYA that identifies multiple modes of activation, that cannot be picked up by current methods of pathway quantification (Pagoda2 for example).

Overall, these two novel analyses illustrate how MAYA brings additional information that could not have been found with traditional approaches, either with GSEA or using average activity score of pathways.

Figure 3R13: Overlaid jitter and boxplot representation of Pearson correlation coefficient between MAYA modes with Hallmark on the ovary dataset with average expression of the corresponding pathways, grouped by modes.

Are all these tumors driven by KRAS or estrogen response?

In fact cancer cells show enrichment in KRAS_SIGNALING_DN (DN="down-regulated" genes) while macrophages are enriched in KRAS_SIGNALING_UP.

Do the authors suggest that cancer cells down-regulate KRAS signaling while macrophages up-regulate this pathway?

We thank the reviewer for bringing to us the misleading results presented in Fig. 5b, where we only display the top 5 specific modes for the four main populations in the dataset. What this figure did not show is that the first mode of the 'KRAS signaling up' gene list is also active in tumor cells (Fig. 3R15). Regarding the 'KRAS signaling down' list, we do detect the specific expression of a subset of genes in epithelial cells, but it does not make much sense to compute the activity of genes that are under-expressed. To ease result interpretation, we decided to remove from MAYA's activity matrix, gene lists that correspond to negative markers (n=2, HALLMARK_UV_RESPONSE_DN and HALLMARK_KRAS_SIGNALING_DN). There are no such lists in the KEGG and REACTOME databases that are used in our analyses.

Figure 3R15: Heatmap of scaled gene expression in the ovary dataset for top10 contributing genes for the modes of Hallmark KRAS signaling up, ordered by decreasing contribution. Author annotation is indicated above heatmap.

Also “It notably detects a cell-type specific activation of complement genes in macrophages.” Why is this notable? Is this proven or have a demonstrated role in these tumors?

As raised by the reviewer, the fact that we detect a mode of activation of the complement pathway specific to macrophages is not notable from a biological perspective in the context of ovarian cancer but is rather a sanity check that we find within MAYA top specific modes well-known and potentially expected pathways. We have modified the text line 236.

“MAYA identifies common modes of activation across tumor cells, which could be compared to such tumor meta-programs” But no comparison is provided; so how to interpret these “modes of activation”?

Following the reviewer’s comment, we have now computed ‘*metaprograms*’ as proposed by Gavish et al. (see Method section, lines 608-613) and compared them to MAYA modes for the ovary dataset.

We identified eight *metaprograms* shared by at least three patients (Fig. 3R16a). However, unlike MAYA modes, *metaprograms* are not directly interpretable and the authors suggest performing functional enrichment of the *metaprograms* to associate them with known pathways (Fig. 3R16b). To compare *metaprograms* and MAYA modes, we computed the percentage of overlap of the top20 contributing genes of MAYA modes presented in Fig.5 and Supplementary Fig. 5 with each of the *metaprograms* (Fig. 3R16c). We noticed that *metaprograms* overlap MAYA modes, but that both approaches are complementary: while MAYA interrogates existing databases to identify modes of pathway activation across tumor cells, NMF extracts patient specific programs that can then be grouped and interpreted. These results were included in the manuscript, lines 268-276 and 281-283, Supplementary Fig. 5.

Figure 3R16: Comparing metaprograms and MAYA modes (a) Heatmap representation of the similarity of robust NMF programs, found in epithelial cancer cells for each patient individually, in the ovary dataset. Similarity is assessed using a Jaccard similarity index of programs, based on their top 50 genes. Main metaprograms are identified as groups of similar programs and circled in black and are characterized by the union of top 50 genes of each contributing program. (b) Barplot displaying the Hallmark pathways enriched in NMF metaprograms. X-axis corresponds to $-\log_{10}$ adjusted p-values. (c) Heatmap displaying the percentage of the top20 contributing genes of MAYA modes with each metaprogram. Only MAYA modes with a percentage of overlap superior to 15% with one metaprogram were kept for clarity.

9) “MAYA focuses on cell identity by looking only at genes considered as markers, it does not detect the variations between patients driven by other sets of genes that are not related to cell type identity and that lead to the formation of different clusters in a classical gene-based analysis.” A fair comparison would be to do clustering using the same set of cell type discriminating genes. The differences shown might be not due to MAYA, but to the list of genes provided to it as input.

Following the reviewer’s suggestion, we ran a gene-based analysis on the larynx and pancreas datasets using only PanglaoDB genes instead of the $n \sim 2,000$ most variable ones (‘GeneBased2’ approach, Fig. 3R17). As for the initial gene-based method, we observe patient or technology specific clusters, showing that it is not the provided gene list but rather MAYA that enables cross-patient cell type identification.

Figure 3R17: (a) Overlaid jitter and boxplot representation of Shannon Diversity Index (SDI), for clusters derived from classical gene-based dimensionality reduction, gene-based dimensionality reduction using PanglaoDB genes only (gene-based2), Harmony dimensionality reduction, SeuratCCA and MAYA activity matrix of the larynx dataset. (b,c) UMAP representation of the larynx dataset using gene-based dimensionality reduction using PanglaoDB genes only, cells are colored according to (b) cell type or to (c) patient. (d) Overlaid jitter and boxplot representation of Shannon Diversity Index (SDI), for clusters derived from classical gene-based dimensionality reduction, gene-based dimensionality reduction using PanglaoDB genes only (gene-based2), Harmony dimensionality reduction, SeuratCCA and MAYA activity matrix of the pancreas dataset. (e,f) UMAP representation of the pancreas dataset using gene-based dimensionality reduction using PanglaoDB genes only, cells are colored according to (e) annotation or to (f) technology.

Minor comments:

- In the Kidney dataset, when matching labels for PanglaoDB “T helper cells”, those are CD4 and not CD8 T cells. Same for “Regulatory T cells” for the colon dataset. All the other categories which are merged are not regulatory T cells.

We thank the reviewer for pointing out some inconsistencies in the reference labels that we chose to compute cell type prediction accuracy for the kidney and colon datasets. Following the reviewer’s recommendation, we have removed CD4 T cells related labels from the possible labels for kidney CD8 T cells, and CD8 T cells related labels from the possible labels for colon regulatory T cells. However, given the overlap between the markers for immune cell types, we find satisfactory enough to annotate CD8 T cells or regulatory T cells as T cells or T memory cells. Reference labels were modified in the manuscript lines 644-651.

- we find confusing the use of “oncogenic” to describe tumor/cancer datasets

We have replaced “oncogenic” by “cancer” in the manuscript.

- the use of “multi-modal” can be misleading as this term is broadly used in the single-cell biology field for multi-omics assays. Consider alternatives, eg multi-modular

MAYA corresponds precisely to the study of the different modes of pathway activation, and in this sense, we do not feel it appropriate to change the term “multi-modal” to another one. In addition, for multiple omics measurement, we have the feeling that multi-omics is the most widely used term.

- line 88: However, not

We have corrected this in the manuscript line 89.

REVIEWER COMMENTS

Reviewer #1 (Remarks to the Author):

The authors have successfully addressed all of my concerns. I have no other comments.

Reviewer #3 (Remarks to the Author):

MAYA is a bioinformatics tool that increases the granularity of pathway analysis of scRNA-seq data. It does so by detecting subsets of genes within the input gene signatures ("modes of activation") which show bimodal expression across cell populations in the dataset. This can be a useful exploratory tool in the field.

In the revised manuscript the authors have clarified multiple points. However, we believe important concerns are still not successfully addressed:

previous point 1): there is no sufficient evidence for the biological relevance of the discovered "modes of activation". By splitting pathway gene sets into multiple subgroups, it is expected by design that MAYA will identify more hits than a standard GSEA (e.g. multiple possible modes for each GSEA signature). However, what is not obvious is that the discovered activation modes are biologically relevant.

Some examples pathway modes of activation provided in the paper:

WNT: Can the authors provide evidence that these two "WNT modes of activation" are supported by external evidence? Is there a statistically significant enrichment of MAYA modes with experimentally validated 'canonical' and 'non-canonical' WNT signaling pathways?

EMT: Can the authors show that this "tumor-specific" EMT module, and not the original one, is specific to cancer cells in external datasets?

What's the meaning of the macrophage-specific and CAF-specific EMT signatures?

Without robust validation, it may be important to underscore that these "modes of activation" should be interpreted with skepticism, as there is no guarantee for their biological relevance.

previous point 2) on generalizability and reproducibility of the modes of activation. The authors conducted an experiment to address this issue in Figure 3R4, where they derived MAYA modes of EMT and coagulation in 3 datasets. However, quantitative data showing that modes found in different datasets are highly similar are lacking. What is the activity score correlation of each mode derived in one dataset and evaluated in the remaining datasets? If activity scores derived from one dataset are evaluated in the others, would they display a strong cell type specificity?

It would also be critical to show reproducibility for all MoA highlighted in the manuscript, such as the WNT canonical/non-canonical modes, the estrogen response early pathway, and the KRAS signaling.

previous point 3) Our concern is related not to the variance explained by the PCA components but to the variance explained by each gene in the original gene expression space. Many genes show minimal variation across cells (e.g. housekeeping genes that can be highly expressed in all cells, or very lowly expressed genes, below the limit of detection). What happens with these genes when MAYA re-scales their noisy expression values? Would they contribute as strongly to PCA components as actual highly variable genes?

RESPONSE TO REVIEWERS' COMMENTS

Reviewer 3

previous point 1): there is no sufficient evidence for the biological relevance of the discovered “modes of activation”. By splitting pathway gene sets into multiple subgroups, it is expected by design that MAYA will identify more hits than a standard GSEA (e.g. multiple possible modes for each GSEA signature). However, what is not obvious is that the discovered activation modes are biologically relevant.

Some examples pathway modes of activation provided in the paper:

WNT: Can the authors provide evidence that these two "WNT modes of activation" are supported by external evidence? Is there a statistically significant enrichment of MAYA modes with experimentally validated 'canonical' and 'non-canonical' WNT signaling pathways?

Wnt signaling is a complex pathway in which β -catenin is typically viewed as a central mediator. Since the 90s, at least three Wnt-mediated pathways have been proposed that function independently of β -catenin (for review see Zhan et al., *Oncogene* 2016). These different pathways and related publications, that support them, are summarized in the KEGG database (<https://www.genome.jp/pathway/hsa04310>, see Figure 3RR1). The Wnt pathway mediated by β -catenin is referred to as the “canonical Wnt pathway”, while the three others have specific denominations based on their members. One pathway involves activation of calcium/calmodulin-dependent kinase II (CamKII), protein kinase C (PKC) and phosphatase CaN (PPP3CC), and is referred to in the literature as the “Wnt Ca²⁺ pathway” (for review see Kohn and Moon 2005). This non-canonical pathway is activated by the ligand WNT5A (Slusarski et al., *Dev Biology* 1997 / Anakwe et al., *Development* 2003).

With MAYA, we have detected two modes of WNT pathway activation (see Fig. 3RR1): one, solely containing genes of the canonical pathway (in pink), and driven potentially by the canonical ligand WNT7A, and a second mode (in blue) containing genes implicated in both canonical and Wnt Ca²⁺ signaling, in particular the ligand WNT5A. Mode 2 – specific to CAFs and mesothelial cells – indicates the cells might be using two Wnt pathways in contrast to tumor cells (mode 1) using solely the canonical pathway. Altogether existing literature confirms the biological relevance of the gene lists detected by MAYA.

We have now included additional evidence supporting the biological relevance of MAYA modes in the manuscript lines 264-271 with references 60 to 63.

Figure 3RR1: Visualization of the KEGG WNT pathway. Top10 contributing genes to the two MAYA modes identified in the ovary dataset are highlighted.

EMT: Can the authors show that this "tumor-specific" EMT module, and not the original one, is specific to cancer cells in external datasets? What's the meaning of the macrophage-specific and CAF-specific EMT signatures? Without robust validation, it may be important to underscore that these "modes of activation" should be interpreted with skepticism, as there is no guarantee for their biological relevance.

Following the reviewer's recommendations, we compared our tumor-specific EMT module to the original Hallmark EMT module (Figure 3RR2) in the three presented datasets and an additional external one (Gavish et al., 2022 – prostate cancer). Our tumor-specific EMT module corresponds to the common set of genes detected by MAYA in the top30 contributing genes of the tumor EMT mode for the ovary, lung and breast datasets. We show that the initial Hallmark EMT module is specific to CAF, whereas our tumor EMT module is significantly higher in tumor cells compared in all datasets, definitively showing the ability of MAYA to deconvolute pathway analysis.

The CAF and macrophage-specific signatures correspond to genes that are expressed respectively in CAF and macrophages and associated to an EMT process, implying that they are the genes in these cells that could be driving a mesenchymal-like state. The genes found in each mode by MAYA, have previously been shown to indeed take part in EMT, supporting the biological relevance of MAYA deconvolution. We had already cited all these references in our last version of the manuscript:

"In CAFs, MAYA detects EMT as driven mainly by genes encoding proteins from the extracellular matrix (ECM) including collagens, which have been shown to promote EMT in the tumor microenvironment

directly⁴⁷ or by increasing the ECM stiffness^{48,49}. A third mode of EMT, characterized by the expression of the gene *SPP1*, is found in macrophages; macrophages have indeed been shown to be involved in EMT induction in various types of cancer⁵⁰⁻⁵³.”

Figure 3RR2: Comparison of tumor-specific EMT module found by MAYA and initial Hallmark EMT module.

Previous point 2) on generalizability and reproducibility of the modes of activation. The authors conducted an experiment to address this issue in Figure 3R4, where they derived MAYA modes of EMT and coagulation in 3 datasets. However, quantitative data showing that modes found in different datasets are highly similar are lacking. What is the activity score correlation of each mode derived in one dataset and evaluated in the remaining datasets? If activity scores derived from one dataset are evaluated in the others, would they display a strong cell type specificity? It would also be critical to show reproducibility for all MoA highlighted in the manuscript, such as the WNT canonical/non-canonical modes, the estrogen response early pathway, and the KRAS signaling.

We have now evaluated the significance of the overlap between the different tumor_EMT modules found by MAYA in various cancer types, and there are all significant (breast/lung $pval=3e-5$, breast/ovary $pval=1e-3$, lung/ovary $pval=4e-6$). We have now also correlated scores obtained with signatures extracted from different datasets (see below examples for the lung dataset) for both the CAF and tumor EMT modes. We observe strong Pearson’s correlation coefficient between the scores computed from different signatures/datasets and a high cell type specificity (Fig. 3RR3). We have included part of these results in the manuscript (Supplementary Fig. 5).

Figure 3RR3: Scatter plots comparing for each cell in the lung dataset the activity scores of the CAF (top) and tumor cell (bottom) EMT signatures derived from the lung, ovary, and breast datasets. The dark straight line represents $y=x$ and Pearson's correlation coefficient is displayed above each plot.

previous point 3) Our concern is related not to the variance explained by the PCA components but to the variance explained by each gene in the original gene expression space. Many genes show minimal variation across cells (e.g. housekeeping genes that can be highly expressed in all cells, or very lowly expressed genes, below the limit of detection). What happens with these genes when MAYA re-scales their noisy expression values? Would they contribute as strongly to PCA components as actual highly variable genes?

To answer the reviewer's question, we have now compared the number of housekeeping or lowly expressed genes that contribute to MAYA modes when scaling or not gene expression before running PCA. To do so, we have tagged in Hallmark genes lists *housekeeping* genes (using the list provided by Eisenberg et al. (Trends Genet, 2011) and *lowly expressed* (genes expressed in less than 5% of the dataset). We found no major differences in gene numbers (Fig. 3RR3): there are actually slightly fewer housekeeping genes in MAYA modes when scaling (5 vs 4 in median) but slightly more lowly expressed genes (median of 0 each, with a maximum of four lowly expressed genes in the top20 contributors when scaling). Altogether, we demonstrate that scaling does not increase noise in MAYA modes. As a reminder, we had decided to keep both options available to the user.

Figure 3RR4: Boxplots representing the number of housekeeping genes (resp. lowly expressed genes) in the top20 most contributing genes to each MAYA mode evaluated in the kidney dataset with Hallmark pathways, before filtering on percentage of activated cells or maximum gene contribution.

Additionally, we now provide examples comparing the contribution of genes with and without scaling for several MAYA modes, highlighting housekeeping and lowly expressed genes (Fig. 3RR4). We show that housekeeping and lowly expressed genes are not more affected than the rest of the genes by scaling.

Figure 3RR5: Scatter plots displaying gene contribution to modes 1 of Hallmark Allograft rejection and TNFA signaling pathways in the kidney dataset, with or without scaling prior to PCA. Housekeeping and lowly expressed genes are highlighted.

REVIEWERS' COMMENTS

Reviewer #3 (Remarks to the Author):

Our comments have been correctly addressed